# Multiplexed ddPCR-amplicon sequencing reveals isolated *Plasmodium falciparum* populations amenable to local elimination in Zanzibar, Tanzania

Aurel Holzschuh [1,2] ✉, Anita Lerch [1], Inna Gerlovina[3], Bakar S. Fakih[2,4,5], Abdul-wahid H. Al-mafazy[6], Erik J. Reaves[7], Abdullah Ali[8], Faiza Abbas[8], Mohamed Haji Ali[8], Mohamed Ali Ali[8], Manuel W. Hetzel [2,4], Joshua Yukich [9] & Cristian Koepfli [1] ✉

Zanzibar has made significant progress toward malaria elimination, but recent stagnation requires novel approaches. We developed a highly multiplexed droplet digital PCR (ddPCR)-based amplicon sequencing method targeting 35 microhaplotypes and drug-resistance loci, and successfully sequenced 290 samples from five districts covering both main islands. Here, we elucidate fine-scale *Plasmodium falciparum* population structure and infer relatedness and connectivity of infections using an identity-by-descent (IBD) approach. Despite high genetic diversity, we observe pronounced fine-scale spatial and temporal parasite genetic structure. Clusters of near-clonal infections on Pemba indicate persistent local transmission with limited parasite importation, presenting an opportunity for local elimination efforts. Furthermore, we observe an admixed parasite population on Unguja and detect a substantial fraction (2.9%) of significantly related infection pairs between Zanzibar and the mainland, suggesting recent importation. Our study provides a high-resolution view of parasite genetic structure across the Zanzibar archipelago and provides actionable insights for prioritizing malaria elimination efforts.

Despite the intensification of control interventions and renewed efforts to eliminate malaria, it remains one of the world's most widespread and deadly infectious diseases, causing 247 million cases and killing more than 600,000 people in 2021[1]. Zanzibar, an archipelago and semi-autonomous region of the United Republic of Tanzania with its two main islands, Unguja and Pemba, has made great progress towards elimination of malaria in the last two decades. Malaria prevalence is now estimated to be less than 5% by PCR[2], mainly as a result of the continued use of long-lasting insecticidal nets (LLIN), indoor residual spraying (IRS), and artemisinin-based combination therapies (ACTs)[3].

Since 2012, the Zanzibar Malaria Elimination Program (ZAMEP) has implemented reactive case detection (RACD) to better target residual foci of transmission[4]. RACD includes screening by rapid

[1]Department of Biological Sciences, Eck Institute for Global Health, University of Notre Dame, Indiana, IN, USA. [2]Swiss Tropical and Public Health Institute, Allschwil, Switzerland. [3]EPPIcenter Research Program, Division of HIV, ID and Global Medicine, Department of Medicine, University of California, San Francisco, CA, USA. [4]University of Basel, Basel, Switzerland. [5]Ifakara Health Institute, Dar es Salaam, United Republic of Tanzania. [6]Research Triangle Institute (RTI) International, Zanzibar, United Republic of Tanzania. [7]U.S. Centers for Disease Control and Prevention, President's Malaria Initiative, Dar es Salaam, United Republic of Tanzania. [8]Zanzibar Malaria Elimination Programme, Zanzibar, United Republic of Tanzania. [9]School of Public Health and Tropical Medicine, Tulane University, New Orleans, LA, USA. ✉e-mail: aholzsch@nd.edu; ckoepfli@nd.edu

diagnostic test (RDT) and treatment of household members of passively detected index cases at health facilities. However, despite considerable efforts, malaria elimination in Zanzibar remains elusive. There are likely several reasons for this: firstly, despite the increasingly focal nature of residual transmission in Zanzibar[3,5–7], there is a substantial reservoir of asymptomatic infections in the community[3]. Many of these infections are low-density below the detection limit of RDTs[2,8,9], making it challenging to identify and treat them. Secondly, despite the implementation of strong vector control measures, there is ongoing local transmission due to residual vector capacity[3]. Lastly, Zanzibar is highly connected to mainland Tanzania where malaria transmission remains substantially higher in certain areas[10–12]. Thus, even if local transmission has been reduced to very low levels, parasite importation through human travel might be a concern and obstacle to local elimination as long as the environment remains receptive[13].

Achieving elimination of malaria in Zanzibar requires a better understanding of why and where importation occurs, as well as the relative contribution of importation to sustaining local parasite populations. Malaria parasite genetic data provides information that is complementary to clinical and epidemiological data[14]. It can support the design of effective interventions by monitoring the spread of drug resistance markers[15–17], assessing transmission intensity[7,18], identifying foci of sustained transmission (i.e., sources and sinks of infections)[19,20], estimating connectivity between parasite populations[21], and distinguishing local from imported cases[19]. Parasite genetic data can identify genetically isolated parasite populations with limited gene flow between them, which might be preferred targets for local elimination by targeted control measures[22]. Further, it is crucial to monitor markers of malaria drug resistance. There is increasing evidence of parasites with markers of artemisinin partial resistance circulating in sub-Saharan Africa, including mainland Tanzania, Rwanda and Uganda, threatening the long-term effectiveness of ACT regimens, the current first-line treatment in Zanzibar[11,23–26].

Novel amplicon sequencing-based genotyping methods allow typing of parasites at high resolution and sensitivity and provide a highly detailed picture of the parasite population structure[27,28]. Targeted amplicon sequencing (AmpSeq) of the most informative regions in the genome allows for deep and consistent sequence coverage[29]. AmpSeq offers very high detectability of minority clones as low as 1% frequency in polyclonal infections[29], which are frequent even in pre-elimination settings[20,30]. By targeting short, highly diverse microhaplotype loci that contain multiple SNPs and exhibit multiallelic rather than biallelic diversity, discriminatory power can be increased[31–33]. To date, many studies utilizing this method have only targeted one or a few genomic loci to gain information on diversity of infections, drug resistance, or selection[29,34–36]. Only recently, there have been efforts to extend these methods to large multiplexed AmpSeq panels covering numerous genetically diverse loci (e.g., microhaplotypes) and markers of drug-resistance, as pioneered by Tessema et al.[32] and others[15,33]. Increasing the number of diverse loci can provide a higher resolution comparison of infections at the population level as well as pairwise relatedness inference at the individual level[15,32,33].

Various genomic indices yield information relevant for malaria genomic epidemiology. Genetic relatedness is a recombination-based measure of recent shared ancestry[37]. It ranges from zero for completely unrelated individuals to one for identical clones. Identity-by-descent (IBD)-based methods allow estimating the probability that two alleles at a given locus in the genome are identical due to descent[38]. This provides a metric at finer spatiotemporal scales (10 s to 100 s of kilometers; weeks to months) for characterizing changes and measuring connectivity in parasite populations than classical population genetic measures of population diversity and divergence (e.g., $\pi$ or $F_{ST}$)[14,21], on timescales relevant to malaria elimination efforts[39]. While many IBD-based methods use whole genome sequencing (WGS) data

and are limited to monoclonal infections or to biallelic loci[40,41], a recently developed approach allows to infer the level of shared common ancestry (i.e., relatedness) between polyclonal infections from unphased multiallelic data such as microhaplotype data generated by AmpSeq[42]. Measures such as population genetic diversity, linkage disequilibrium, or multiplicity of infection (MOI) might be useful as surrogate markers of changes in transmission intensity[7,18,30].

Previous studies that have examined the population structure of *P. falciparum* in island settings potentially closely linked to mainland areas with higher transmission analyzed a limited number of samples and typed only few genomic markers[43,44]. Limited genomic data is available from the Zanzibar archipelago and its parasite connectivity to the mainland[45]. It is unclear whether additional infections identified through RACD are directly related to clinical index cases. A thorough genetic characterization of circulating parasites on the archipelago is crucial to elucidate travel patterns, and to identify within-country and cross-border parasite connectivity.

We developed a novel high-throughput multiplexed AmpSeq method targeting highly diverse microhaplotypes and drug resistance loci[29,32,35] to characterize the parasite population structure and map antimalarial resistance profiles in Zanzibar, and define relatedness of parasites. The assay was optimized for high sensitivity to allow typing of asymptomatic, low-density samples collected as dried blood spot (DBS) on filter papers, which are a practical sample type for large-scale epidemiological studies. Here, we utilize multiplexed AmpSeq designed to quantify differentiation and relatedness between samples. We evaluate the parasite population diversity in Zanzibar and use this information to assess the relatedness between clinical index cases and the asymptomatic cases surrounding them identified by RACD, and to quantify relatedness of and connectivity between parasites on Zanzibar and mainland Tanzania and Kenya.

## Results

### Performance of multiplexed AmpSeq on cultured isolates

We developed and optimized a novel 2-step multiplex ddPCR-based assay for even amplification and detection of minority clones and validated the assay on a range of 3D7 whole blood and DBS controls. For control samples, the assay generated a high number of reads per sample and per locus after trimming low quality reads for both, whole blood samples and DBS samples (Supplementary Table 1). The assay achieved even coverage across a wide range of parasite densities in whole blood (Supplementary Fig. 1A), with 85% to 100% of the 35 loci amplified in samples down to 10 parasites/μL (Supplementary Fig. 1B). For DBS, reads were also evenly distributed among the 35 loci, although more variation was observed compared to whole blood samples (Supplementary Fig. 1C). 81 to 94% of loci were successfully amplified in samples with as few as10 parasites/μL (Supplementary Fig. 1D). In all tests using culture strains, no false alleles were detected, resulting in a precision (defined as TP/(TP + FP)) of 3D7 microhaplotype calls of 100% for all parasite densities evaluated.

We determined the limit of detecting the minority clone at lower parasite densities using mock samples containing both 3D7 and FCB or 3D7 and HB3 strains at various ratios and parasite densities (Supplementary Table 2). At 40% frequency, the minority clone was detected at 100% of dimorphic loci down to 100 parasites/μL, and at 66.7–77.8% of positions when consisting of 20% of the mixture. At 10 parasites/μL, minority clone detection was still high at 40% (63.0–92.6%) and 20% frequency (48.1–66.7%). At very low frequency of 2%, minority clones were detected at approximately 52.4–71.4% of loci that were discordant between the two strains. Detection at 1% frequency was 28.6–42.9%. Lowering the detection limit (i.e., cut-off) from 1 to 0.1% improved recovery of minority clones, especially at 2% and 1% frequency. However, a cut-off should be applied to

prevent false-positive haplotype calls[29]. More even amplification of markers using the droplet-based approach was observed. The fold-difference between the highest number of reads and lowest number of reads per sample was lower when using the ddPCR compared to conventional PCR (Supplementary Table 3). Further, the proportion of reads lost after trimming of low-quality reads was lower when using ddPCR (at 5 parasites/µL: 4.1–5.1% for ddPCR and 10.2–11.0% for conventional PCR). Coverage of markers was identical between the two methods.

**Multiplexed AmpSeq of field samples from Zanzibar**
We then applied the method to all 518 positive *P. falciparum* DBS samples (geometric mean density 4.87 [IQR: 0.07–175] parasites/µL, Supplementary Fig. 2) from Zanzibar collected between 2017 and 2018. All samples were sequenced in duplicate. A total of 290 (56%) DBS samples with data in ≥10 loci for both replicates and a minimum of 10 reads per marker were included in further analyses (Fig. 1a, Supplementary Data 1 and 2). These samples are representative of all 5 districts from which the original sample set was collected, with no clear differences in the proportion of samples from the same vs different households between Unguja and Pemba (Supplementary Tables 4 and 5). The median number of reads per sample in those 290 samples was 549,618 (IQR: 235,051–1,015,204). There was a strong correlation ($R^2 = 0.72$; $P < 0.001$) between the number of markers successfully genotyped and parasite density, with samples genotyped at ≥10 loci having significantly higher geometric mean parasite densities compared to samples genotyped at 9 loci or less (137.36 parasites/µL vs. 0.07 parasites/µL, Welch's *t* test, $P = 0.0028$). At densities ≥5 parasites/µL, 99.2% (241/243) samples had data for ≥10 loci. High

coverage (median 98.6%) in samples with ≥10 parasites/µL was achieved (Fig. 1b). However, the average number of reads per target in the 290 samples with ≥10 loci showed considerable variation among some of the markers (Fig. 1c). All markers were present with ≥10 reads in 80% of samples, except for marker *t01*, which was underrepresented with ≥10 reads in only 57% of samples (Fig. 1d).

**Multiplicity of infection and population diversity in Zanzibar**
The majority (60%) of infections in Zanzibar were polyclonal with a mean MOI of 1.85 (range 1–8) across districts (Fig. 2b, Supplementary Fig. 3A). The genetic diversity of the parasite population was high throughout Zanzibar (median $H_E = 0.75$ [IQR: 0.59–0.88]) with a high number of unique alleles (mean: 20.4 per marker, SD: 22.8 range: 2–102) (Supplementary Fig. 3B). There was a moderate correlation between the expected heterozygosity based on a global dataset of *P. falciparum* genomes and the observed heterozygosity in Zanzibar ($R^2 = 0.37$; $P < 0.001$) (Fig. 2b). Correlation was higher when sub-setting to samples from East Africa or Tanzania only, respectively (Supplementary Fig. 4A, B). Overall, this shows that the microhaplotypes used are highly polymorphic both in Zanzibar and globally. The microhaplotypes were highly diverse in each of the five districts of Zanzibar, though moderately lower in Micheweni (median $H_E$: Micheweni: 0.67, Chake Chake: 0.73, Mkoani: 0.73, Magharibi: 0.76, and Kusini: 0.73) (Fig. 2c). We also investigated heterozygosity and MOI by travel history, year, and island and found significantly higher population diversity and MOI on Unguja than Pemba (MOI 1.95 vs. 1.69, $P = 0.021$; $H_E$ 0.73 vs. 0.70, $P = 0.012$), as well as higher MOI in individuals who reported recent travel (2.13 vs. 1.72, $P = 0.008$) (Supplementary Fig. 5A–F).

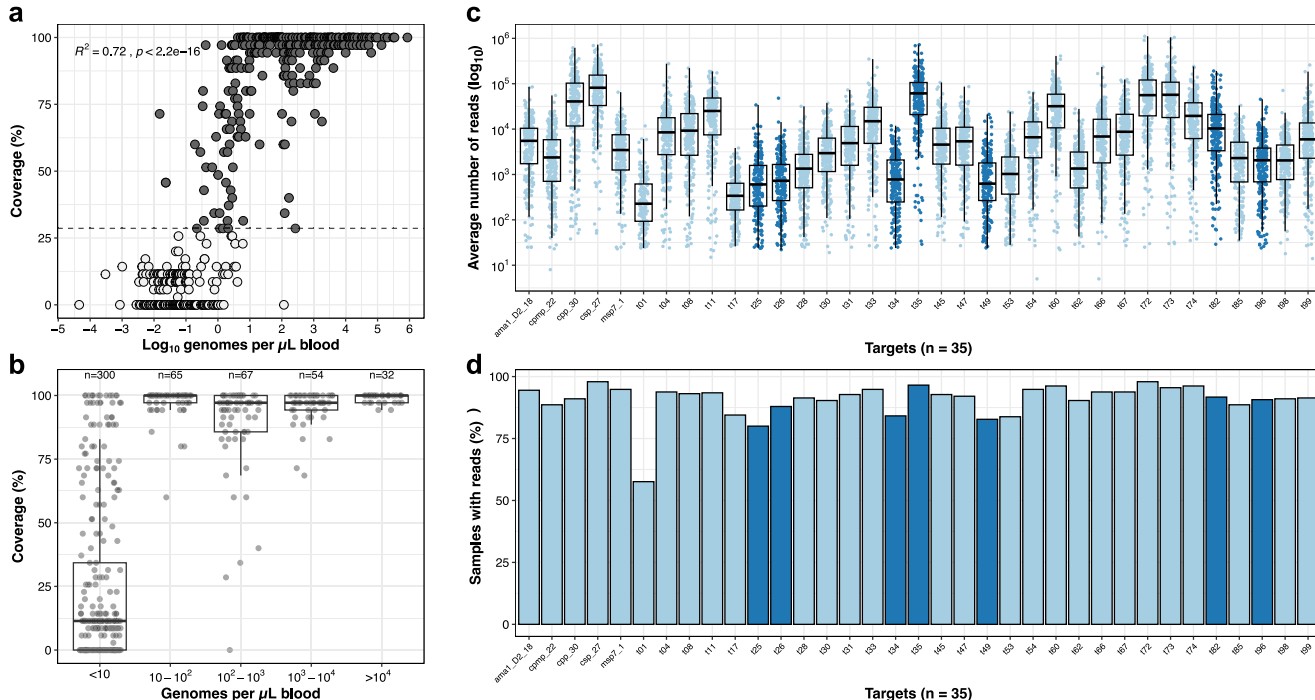

**Fig. 1 | Evenness and coverage of multiplexed amplicon sequencing of microhaplotypes (*n* = 28) and drug resistance loci (*n* = 7). a** Coverage of microhaplotype loci and drug resistance targets by parasite density in 518 DBS samples from Zanzibar. 290 samples with data in ≥10 loci (dashed line) were included for further analyses (colored in black). Correlation between sequencing coverage and parasite density was analyzed using linear regression ($R^2 = 0.72$, $F(1516) = 1306$, $P < 2.2e{-}16$); *P* value is two-sided. **b** Boxplot summarizing the coverage of microhaplotype loci and drug resistance targets by parasite density in 518 DBS samples from Zanzibar. Coverage was determined based on the number of targets with 10 or

more reads (e.g., 100% indicates 35/35 loci with ≥10 reads). Sample sizes for each bin are indicated. The box bounds the IQR divided by the median, and Tukey-style whiskers extend to a maximum of 1.5 × IQR beyond the box. **c** Boxplot showing the average number of reads per target (*n* = 35) per sample (*n* = 290). Colored by microhaplotypes (light blue) and drug resistance loci (dark blue). The box bounds the IQR divided by the median, and Tukey-style whiskers extend to a maximum of 1.5 × IQR beyond the box. Note that the *y*-axis is on a log₁₀-scale. **d** Number of samples (%) of the 290 samples included with ≥10 reads per marker. Colored by microhaplotypes (light blue) and drug resistance loci (dark blue).

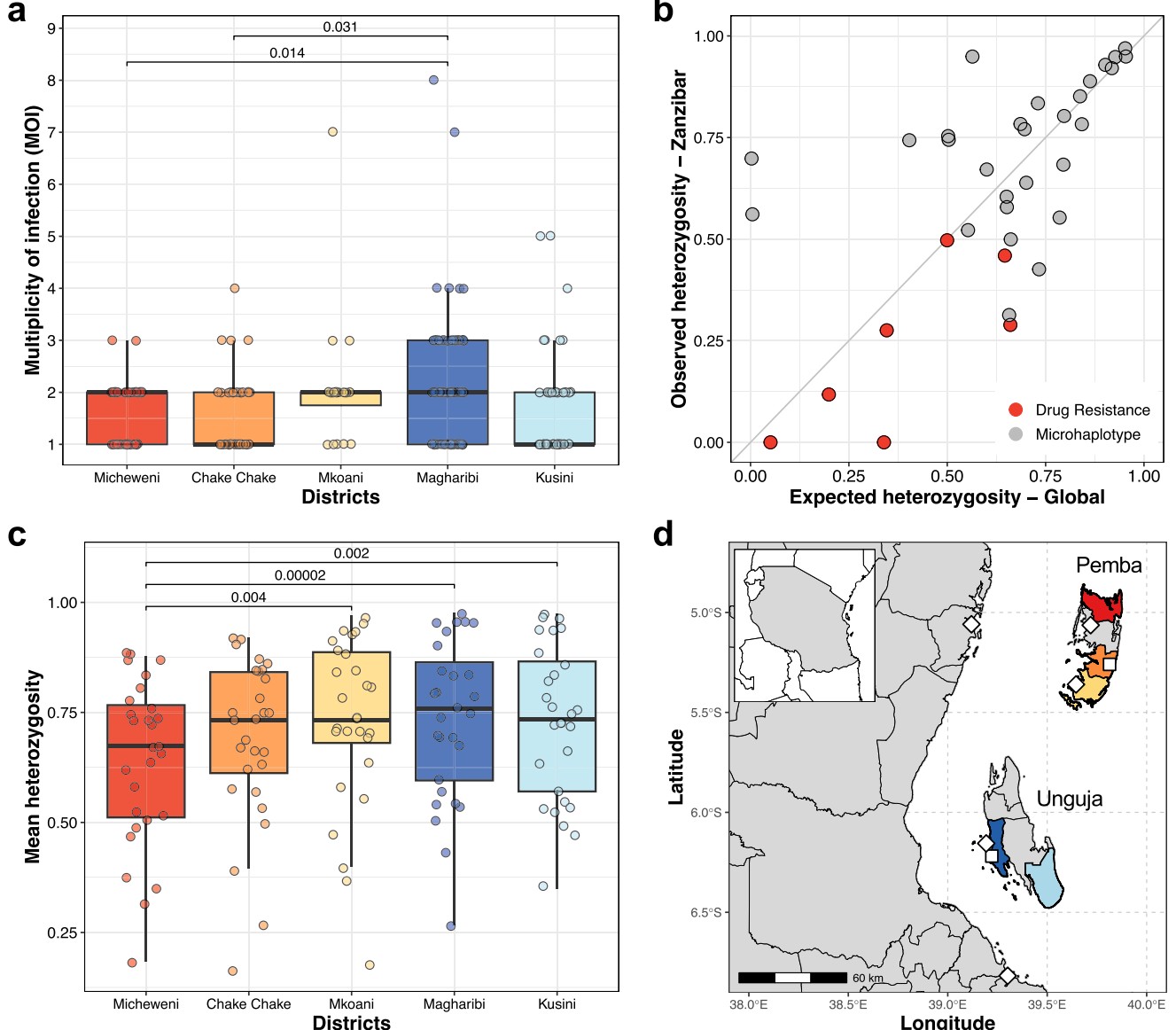

**Fig. 2 | Multiplicity of infection and heterozygosity. a** Multiplicity of infection (MOI) of 290 DBS samples across 5 districts. The box bounds the IQR divided by the median, and Tukey-style whiskers extend to a maximum of 1.5 × IQR beyond the box. Differences between districts were examined using a two-sided *t* test. Bonferroni adjusted *P* values are indicated for significant pairs only. **b** Correlation of expected heterozygosity of microhaplotypes and drug resistance loci in the global population and the observed heterozygosity in the Zanzibar samples was analyzed using linear regression ($R^2 = 0.37$, $F(1, 33) = 19.78$, $P = 9.292e-5$). **c** Distribution of expected heterozygosity of the 28 microhaplotypes across 5 districts in Zanzibar. The box bounds the IQR divided by the median, and Tukey-style whiskers extend to a maximum of 1.5 × IQR beyond the box. Differences between districts were examined using a pairwise two-sided *t* test. Bonferroni adjusted *P* values are indicated for significant pairs only. **d** Map of Zanzibar archipelago with the 5 districts included in the study. Inset map shows the United Republic of Tanzania. Airports (squares) and ferry terminals (diamonds) are highlighted.

## Genetic differentiation and population structure in Zanzibar

Population measures of genetic differentiation ($G''_{ST}$ and *Jost's D*) identified high genetic differentiation of two districts on Pemba, Micheweni and Chake Chake, from the other districts (Supplementary Fig. 6A). The other three districts showed very little differentiation among each other. Similarly, we found higher differentiation with increasing geographic distance between 29 shehias with at least three infections each (Supplementary Fig. 6B).

Population cluster analysis identified three clusters, with most parasites from Micheweni and Chake Chake being assigned to clusters 2 and 3, respectively (Supplementary Fig. 6C). Samples from Mkoani in the south of Pemba and samples from Magharibi and Kusini on Unguja had signs of increased admixture, compared to samples from northern

Pemba. DAPC using the first 23 components explaining 49.8% variation of the original PCA (Supplementary Fig. 7A, B) inferred 3 distinct subpopulations by their district-level origin (Fig. 3, Supplementary Fig. 8A, B), highlighting spatial clustering between districts, particularly separating samples from Micheweni and Chake Chake into more isolated populations. Individuals from Pemba districts Micheweni (8.8%), Chake Chake (16.3%) and Mkoani (15.0%) reported less travel outside of Zanzibar than individuals from Unguja districts, i.e., Magharibi (32.2%) and Kusini (49.0%). Overall, travel outside of Zanzibar was higher on Unguja than on Pemba (37.1% vs. 12.5%, chi-square test, $P < 0.0001$). Similarly, inbreeding determined by mLD was higher in samples from Micheweni and Chake Chake, indicating the circulation of more closely related haplotypes in these populations (Supplementary Table 6).

Overall, this suggests frequent recombination of different parasite clones on both islands, but also locally restricted transmission, in particular, on Pemba.

## Pairwise genetic relatedness of parasites

We estimated the genetic relatedness of all pairs of samples ($n = 41,905$) by pairwise identity by descent (IBD) using *Dcifer*. Expected sharing is 0.50 for full siblings and 0.25 for half-siblings with

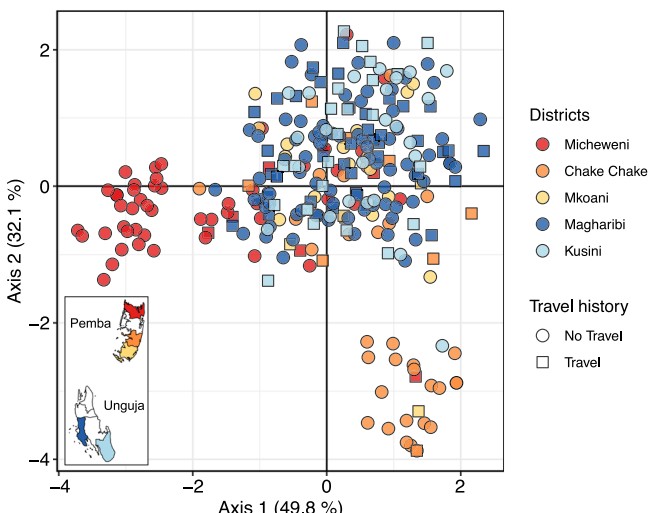

**Fig. 3 | *P. falciparum* population structure across 5 districts in Zanzibar.**
Separation of *P. falciparum* samples by district using discriminatory analysis of principal components (DAPC). The first 23 components of a principal component analysis (PCA) are shown. Each dot represents a sample colored by its geographic origin based on the 5 districts included in the analysis (inset map). Travel history is indicated by circles (no travel) or squares (travel).

unrelated parents[46]. Among all pairs of parasites from Zanzibar, mean IBD was 0.037. We found the overall distribution of pairwise IBD to be heavy-tailed, with the majority of samples being unrelated and a tail of very highly related samples (Fig. 4a). Overall, we found 4.1% (1703) significantly related pairs ($P < 0.05$ by *Dcifer*). We found a decay in genetic relatedness with increasing spatial distance in Zanzibar (Fig. 4b), consistent with the classical pattern expected under isolation by distance[47]. This was also true when removing RACD pairs from the analysis (Supplementary Fig. 9A). Pairs collected on the same day showed higher mean IBD than samples collected further apart (Fig. 4c). However, the big difference in mean IBD from samples collected on the same day and those temporally further apart was mostly driven by RACD samples coming from the same day (Supplementary Fig. 9B). Mean IBD was 3.25-fold higher within RACD clusters compared to between RACD clusters (0.13 vs. 0.04, permutation test, $P < 0.0001$), indicating high relatedness of index cases and secondary cases. We also compared mean IBD and the proportion of related infections on different administrative units, by years, and by travel history (Supplementary Table 7). We found that higher mean IBD and proportion of related infections at the district-level was mostly driven by two districts on Pemba, Micheweni and Chake Chake, with much higher mean IBD and proportion of related infections (Supplementary Fig. 10A, B). When we stratified the districts by year, we found that mean IBD and proportion of related infections in Micheweni and Chake Chake districts were higher in samples from 2017 than 2018 (Supplementary Fig. 10C, D). However, we observed distinct patterns for the two districts. Samples in Micheweni share a high proportion between transmission seasons (mean IBD = 0.101, permutation test, $P < 0.0001$; proportion related = 10.7%, permutation test, $P < 0.0001$), indicating that a proportion of the parasite population is maintained through the dry season and contributes to malaria cases the following season. We also found a 1.3-fold higher mean IBD among non-travelers vs. among travelers to mainland Tanzania and Kenya (0.04 vs. 0.03, permutation test, $P = 0.003$), possibly due to subtle population structure differences between Zanzibar and the mainland.

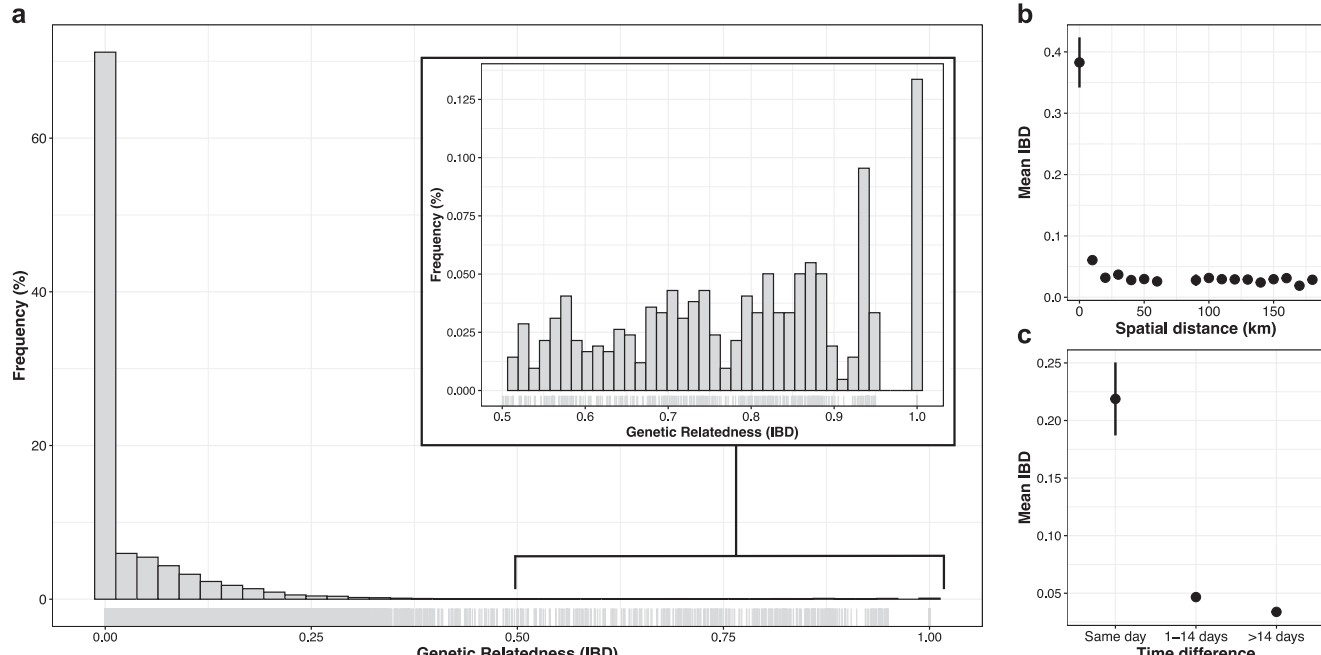

**Fig. 4 | Pairwise genetic relatedness and spatiotemporal patterns of IBD.**
**a** Histogram of pairwise IBD between all samples ($n = 290$ samples; $n = 41,905$ pairwise comparisons), estimated by Dcifer[42]. Inset shows the heavy tail of the distribution, with some pairs of samples having IBD ≥ 0.9. **b** Spatial patterns in IBD among all samples ($n = 290$ samples; $n = 41,905$ pairwise comparisons). Mean IBD

binned by the spatial distance in 10 km intervals. Bin at distance 0 km represents within-household comparisons. Vertical lines indicate 95% confidence intervals.
**c** Temporal patterns in IBD among all samples ($n = 290$ samples; $n = 41,905$ pairwise comparisons). Mean IBD of sample pairs collected on the same day, 1–14 days apart, or >14 days apart. Vertical lines indicate 95% confidence intervals.

Of the 1703 significantly related pairs we found 1335/40,432 (3.3%) pairs between shehias (Supplementary Fig. 11A) and 368/1,473 (24.9%) pairs within shehias (OR = 9.8 [CI: 95% 8.6–11.1], $P < 0.0001$; Supplementary Fig. 11B). Mean IBD among related pairs was significantly higher within shehias than between shehias (0.75 vs. 0.31, permutation test, $P < 0.0001$), with the majority of within shehia related pairs found on Pemba (85.9%; 316/368). Similarly, the odds of finding related infections were significantly higher within RACD clusters than between RACD clusters (46.1% vs. 3.8%; OR = 21.5 [CI 95%: 16.5–28.0], $P < 0.0001$), and within households than between households (46.9% vs. 3.9%; OR = 21.5 [CI: 95% 14.8–31.2], $P < 0.0001$).

## Local transmission and genetic connectivity within Zanzibar

Focusing on the tail of highly related samples, we found 515 sample pairs (1.23%) related at the level of full siblings or closer (IBD ≥ 0.5), including 119 (0.28%) highly related sample pairs (IBD ≥ 0.9). These highly related pairs were found more frequently within the same shehia than between different shehias (Table 1), as well as within the same RACD clusters than in different RACD clusters (Table 1) suggesting the presence of local clonal transmission chains. We found a higher proportion of highly related infections within households than between different households (Table 1), indicating that transmission-related infections cluster within households. This was also reflected by higher mean IBD within than between groups (Table 1). The levels of relatedness among pairs of samples were lowest where one traveled and highest among pairs without any travel.

We grouped all highly related samples and identified 17 distinct clusters that contained between 2 and 19 samples (Fig. 5a). Fourteen clusters contained only samples from the same district, two contained samples from two districts (clusters 'f' and 'n'), and one contained samples from three districts (cluster 'e'), suggesting the presence of local transmission chains. Similarly, most clusters (11/19) contain infections from the same shehia only (Fig. 5b). Of particular interest is the large cluster 'e' that mainly consists of samples from Ndagoni shehia in Chake Chake district (Fig. 5b). The two individuals with infections detected in Micheweni and Mkoani districts had traveled to Ndagoni shehia in Chake Chake, and all infections from Ndagoni were collected 2 weeks after the reported travel. This cluster thus likely represents transmission stemming from an introduced infection.

Fourteen clusters contain samples from the same year only and three clusters (clusters 'a', 'b', and 'e') include samples from both years, spanning periods from 4 months to >1 year (Fig. 5c). This indicates that some local transmission chains are maintained, and a proportion of the parasite population contributes to cases of disease during the next transmission season. Relatedness among the clusters was variable but generally very low, with mean IBD among clusters ranging from 0 to 0.635 (Fig. 5d). Only clusters 'e' and 'f' are more closely related to each other, of note most of these samples originated from the same shehia (Ndagoni). The low relatedness between clusters suggests presence of several independent transmission foci. To understand relatedness at lower relatedness thresholds, we iterated the clustering algorithm across a range of IBD thresholds (Supplementary Figs. 12 and 13).

## Relatedness with mainland Tanzania and Kenya

To investigate connectivity to mainland Tanzania and address the contribution of imported parasites, we estimated genetic relatedness by IBD between all samples from Zanzibar ($n = 290$) and published genomes from mainland Tanzania ($n = 80$) and Kenya ($n = 35$). We specifically chose sites that represent travel destinations reported by the Zanzibari study participants (Fig. 6a). Most pairs were effectively unrelated (median IBD < 0.001), but 1.86% of all pairs were related at the level of half-siblings or closer (IBD ≥ 0.25), including Zanzibar-mainland Tanzania pairs (165/23,200; 0.71%) and Zanzibar-Kenya pairs (66/10,150; 0.65%) (Fig. 6b). Overall, we found 2.9% (976/33,350) significantly related pairs between Zanzibar and mainland Tanzania/Kenya. Mean IBD and the proportion of related infections between Zanzibar-mainland Tanzania and between Zanzibar-Kenya were comparable (Supplementary Table 8). Stratifying by the 5 districts, we find clear differences in their mean IBD and the proportion of related infections to the mainland (Supplementary Fig. 14A, B), corresponding well for districts in Unguja with more frequently reported travel to the mainland, and northern districts of Pemba with less frequently reported travel to the mainland (Supplementary Fig. 14C) and reported travel of any household member of infected individuals (Fig. 7a, b, Supplementary Fig. 14D). The southern district of Pemba, Mkoani, shows higher relatedness but less frequently reported travel to the mainland. However, Mkoani is the main port of entry when traveling from Unguja or Tanga on mainland Tanzania, possibly explaining this

**Table 1 | Proportions of highly related infection pairs (IBD ≥ 0.9) and mean IBD within and between different groups (shehia, RACD cluster, household) and stratified by travel history between pairs of infection**

| Groups | % Highly related sample pairs within (n/N) | % Highly related sample pairs between (n/N) | P value | Mean IBD$_{within}$ (n pairs) | Mean IBD$_{between}$ (n pairs) |
|---|---|---|---|---|---|
| All pairs ($n = 41,905$) | | | | | |
| Shehia | 5.98% (88/1473) | 0.08% (31/40,432) | <2.2e−16 | 0.198 (1473) | 0.031 (40,432) |
| RACD cluster | 17.67% (41/232) | 0.19% (78/41,673) | <2.2e−16 | 0.381 (232) | 0.035 (41,673) |
| Household | 23.89% (27/113) | 0.22% (92/41,792) | <2.2e−16 | 0.407 (113) | 0.036 (41,792) |
| Pairs with no travel ($n = 22,366$) | | | | | |
| Shehia | 6.98% (83/1189) | 0.15% (31/21,177) | <2.2e−16 | 0.227 (1189) | 0.031 (21,177) |
| RACD cluster | 22.4% (37/165) | 0.35% (77/22,201) | <2.2e−16 | 0.457 (165) | 0.038 (22,201) |
| Household | 33.8% (25/74) | 0.40% (89/22,292) | <2.2e−16 | 0.490 (74) | 0.040 (22,292) |
| Pairs where one traveled ($n = 16,536$) | | | | | |
| Shehia | 0.97% (2/206) | 0% (0/16,330) | NA | 0.057 (206) | 0.031 (16,330) |
| RACD cluster | 5.26% (2/38) | 0% (0/16,498) | NA | 0.124 (38) | 0.031 (16,498) |
| Household | 0% (0/13) | 0.01% (2/16,523) | NA | 0.146 (13) | 0.031 (16,523) |
| Pairs where both traveled ($n = 3003$) | | | | | |
| Shehia | 3.85% (3/78) | 0% (0/2925) | NA | 0.135 (78) | 0.034 (2925) |
| RACD cluster | 6.90% (2/29) | 0.03% (1/2974) | <2.2e−16 | 0.281 (29) | 0.034 (2974) |
| Household | 7.69% (2/26) | 0.03% (1/2977) | <2.2e−16 | 0.303 (26) | 0.034 (2977) |

All P values were two-tailed and computed using the chi-squared test for comparing the proportion of highly related samples.

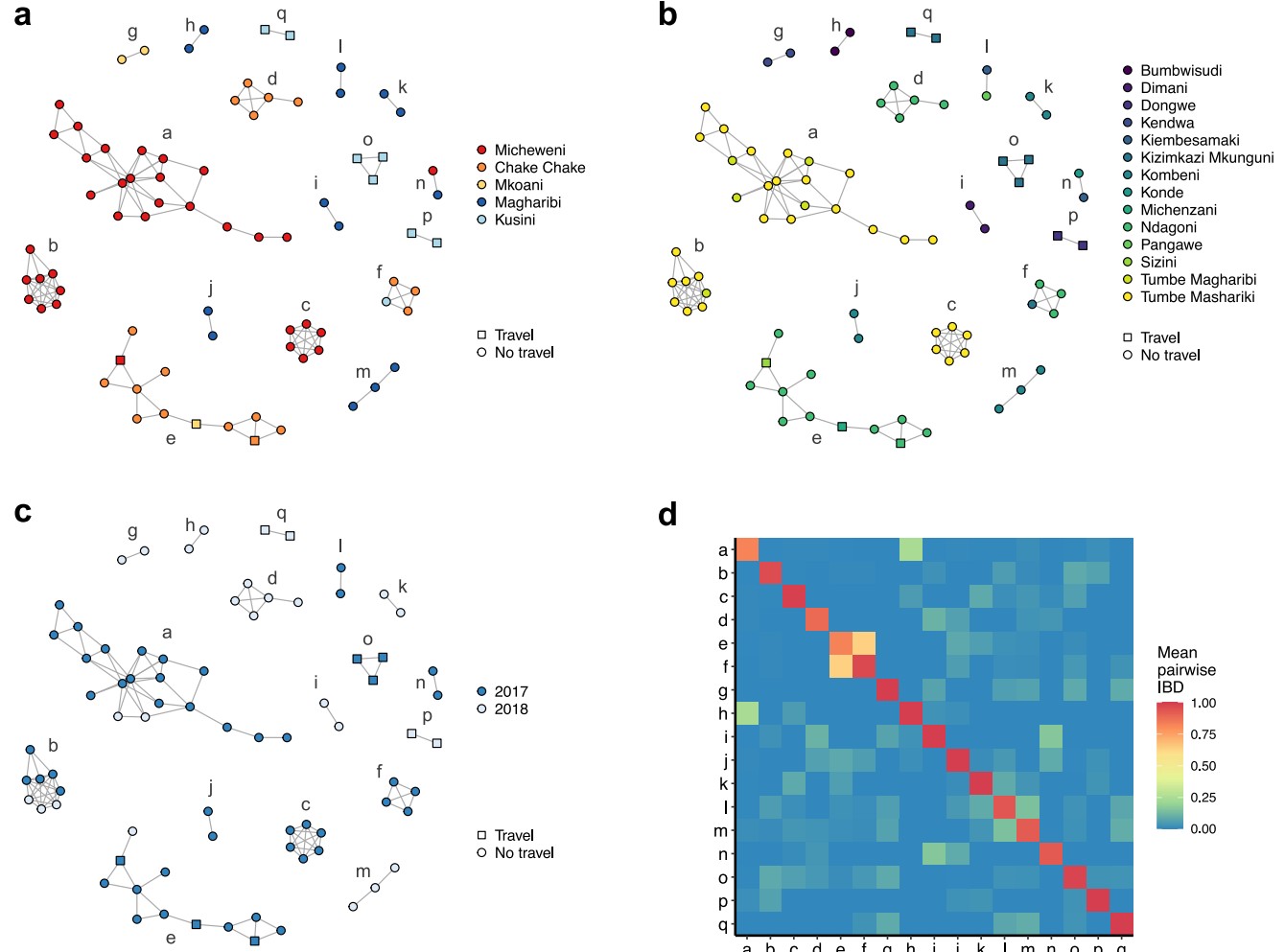

**Fig. 5 | Relatedness network of highly related infection pairs.** Each node identifies a unique infection and edges connecting nodes correspond to IBD ≥ 0.9. Colored according to (**a**), district; (**b**), shehia; and (**c**), year. Travel (square) and no travel (circle) of individuals are indicated. Clusters are labeled with letters. **d** Heatmap showing mean IBD within and between clusters. Mean IBD between clusters ranges from 0 to 0.635 and within clusters from 0.823 to 1.

phenomenon. Mean IBD and the proportion of related infections was significantly higher between the mainland and Unguja than between the mainland and Pemba (mean IBD: 0.025 vs. 0.021, permutation test, $P = 0.0001$; proportion related: 0.034 vs. 0.023, permutation test, $P < 0.0001$; Supplementary Table 8).

Overall, isolates from individuals with recent travel had slightly higher mean IBD with isolates from mainland Tanzania and Kenya than isolates from non-travelers (mean IBD: 0.025 vs. 0.023, permutation test, $P = 0.021$). Three pairs with IBD > 0.5 were identified between three mainland samples and two individuals from Zanzibar (dark-red lines; both from Magharibi district; Supplementary Fig. 15); one of the two individuals reported recent travel to mainland Tanzania. When comparing relatedness of infections among travelers vs. non-travelers stratified by island and district to mainland Tanzania/Kenya, mean IBD was significantly higher in Pemba and two of its districts (Chake Chake and Mkoani) for travelers, but did not differ for Unguja and any of its districts (Supplementary Table 8).

### Global context of parasite haplotypes from Zanzibar
To evaluate the applicability of the microhaplotype panel for geographical attribution of samples and to place Zanzibar samples in the context of global genetic diversity of *P. falciparum*, we performed DAPC using existing genome data from around the globe (Fig. 8a, c). Concordant with existing literature[48], samples could be attributed to three broad clusters corresponding to southeast Asia, east Africa and west Africa, confirming the validity of the microhaplotype panel to distinguish samples by continent of origin, though not by country. Zanzibar and mainland Tanzania samples fell in the east Africa cluster. When including drug resistance loci in the analysis, the separation between east and west Africa became more pronounced and the southeast Asian cluster started to separate into east and west southeast Asia (Supplementary Fig. 16).

Finally, we investigated the relatedness of our Zanzibar samples to countries in east and west Africa. Consistent with the DAPC results, we found that the Zanzibar samples appear to be more closely related to samples from east Africa (i.e., Kenya, Malawi, and Tanzania) and central Africa (i.e., DRC) than to samples from west Africa and southeast Asia (Fig. 8b, Supplementary Table 9). However, relatedness within Zanzibar was higher than between any other country. Previous studies on *P. falciparum* genetic diversity on global scale showed that low IBD levels within sites are predominant in African populations and intermediate/high levels in Southeast Asian populations[48–51]. The same pattern was observed by our panel of markers (Supplementary Table 10).

### Prevalence of drug-resistance markers
We measured the prevalence of drug resistant parasites in Zanzibar, focusing on five markers: *pfdhfr* (codons 50, 51, 59, 108 and 164), *pfdhps* (codons 540, 581 and 613), *pfmdr1* (codons 86 and 184), *pfmdr2*

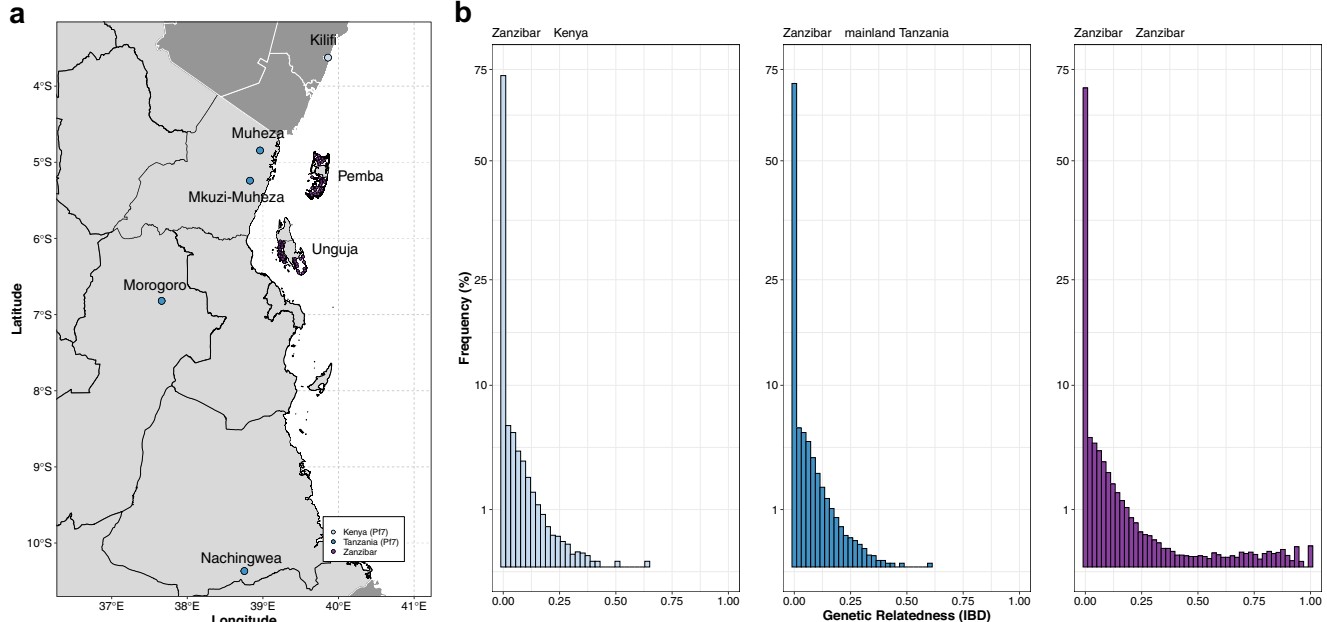

**Fig. 6 | Pairwise genetic relatedness between isolates from Zanzibar and mainland Tanzania and Kenya. a** Location of samples colored by population: purple, Zanzibar; dark blue, published mainland Tanzania *P. falciparum* isolates from MalariaGEN (Pf7)[76]; light blue, published mainland Kenya *P. falciparum* isolates from MalariaGEN (Pf7)[76]. **b** Histogram of pairwise IBD. Pairwise IBD between all samples from Zanzibar (*n* = 290), with mainland Tanzania (*n* = 80), Kenya (*n* = 35) and themselves, estimated by *dcifer*[42]. Note that the y-axis is on square-root scale.

(codons 484 and 492), and *pfk13* (codons 481, 493, 527, 537, 538, 539 and 543) (Table 2). Except for *pfk13*, mutations were found in all genes (Table 2). When stratified by districts, we observed significantly differential frequencies of mutations for *pfdhfr* and *pfdhps* (Supplementary Table 11).

Focusing on 74 Zanzibar samples that were identified as monoclonal infections with complete allele calls for all seven drug resistance loci, we measured the prevalence of multidrug-resistant parasites and explored their spatial distribution (Supplementary Fig. 17A, B). No isolate carried sensitive haplotypes across all five markers analyzed. All isolates carried at least five alleles known to be associated with drug resistance. Most parasites carried drug resistant-associated mutations in the *pfdhps*, *pfdhfr*, and *pfmdr1* genes or *pfdhps* and *pfdhfr* genes. All parasites carried mutations in the *pfdhfr* gene, with the common *pfdhfr* triple mutant (51I-59R-108N) being particularly prevalent (66/74; 89.2%). We also observed a high number of the *pfmdr1* N86-184F haplotype (38/74; 51.4%). We found no obvious spatial pattern for any of the drug resistance alleles.

## Discussion

We have applied a novel, multiplexed ddPCR-amplicon sequencing protocol targeting 28 microhaplotypes loci to provide a high-resolution view of parasite population structure and genetic relatedness of parasites in Zanzibar. We were able to capture information from polyclonal infections (60% of the study population) by using multiallelic loci, and to obtain robust results from low-density asymptomatic infections collected as DBS in the frame of routine RACD procedures Although commonly believed that *P. falciparum* genetic diversity declines with decreasing transmission intensity[18], we found overall high genetic diversity in Zanzibar with only subtle differences between districts, indicative of ongoing transmission and frequent importation. Similarly, a recent study in Eswatini found high parasite diversity despite being considered a low transmission area[52] and diversity might only decline once transmission is at very low levels[53]. The effect of importation needs to be considered, particularly as countries approach elimination, and therefore MOI and genetic

diversity may not be the most appropriate metrics to use as surrogate markers of transmission intensity in Zanzibar and other areas facing similar challenges. In parallel, pronounced population structure and expansion of clonal lineages were observed, mostly on Pemba. The observed population structure matches well with patterns of human movement. The main port of entry to the archipelago is in Zanzibar City, which is bordering Magharibi district on Unguja. Self-reported travel outside of Zanzibar on Unguja was much higher than on Pemba (37.1% vs. 12.5%). In line with the expectation of more frequent parasite importation on Unguja, parasite diversity was higher than on Pemba, and no population structure was observed in those highly admixed parasite populations, suggesting that importation from mainland Tanzania and Kenya play an important role. In contrast, several clonal or near-clonal outbreaks were observed in Micheweni and Chake Chake districts in Pemba, indicative of lower levels of importation from outside. This was further supported by increased genetic differentiation and higher levels of inbreeding in those two districts.

Estimation of pairwise genetic relatedness using an IBD-based approach to infer the degree of shared ancestry between polyclonal infections from unphased multiallelic data[42] allowed for the incorporation of data from all parasites detected in infections. Despite high genetic diversity, we were able to extract useful signals of recent transmission created by recombination and co-transmission of multiple parasites and elucidate fine-scale population structure in Zanzibar, providing evidence of ongoing local transmission. Many malaria cases observed in two districts (Micheweni and Chake Chake) on Pemba appeared to be due to local transmission, evidenced by the strong fine-scale spatial structure in the genetic data. This was further supported by classical measures of genetic differentiation (i.e., DAPC, $G''_{ST}$ and STRUCTURE). Our genomic data corroborated previous spatio-temporal analyses on routinely collected data in Zanzibar from 2015 to 2020 that identified Micheweni district as a hotspot area with greater and more consistent local transmission than other districts[54,55]. We showed that parasites from individuals living closer together (i.e., same household or shehia) were more highly related than more distant infection pairs. Infection pairs from within the same RACD clusters

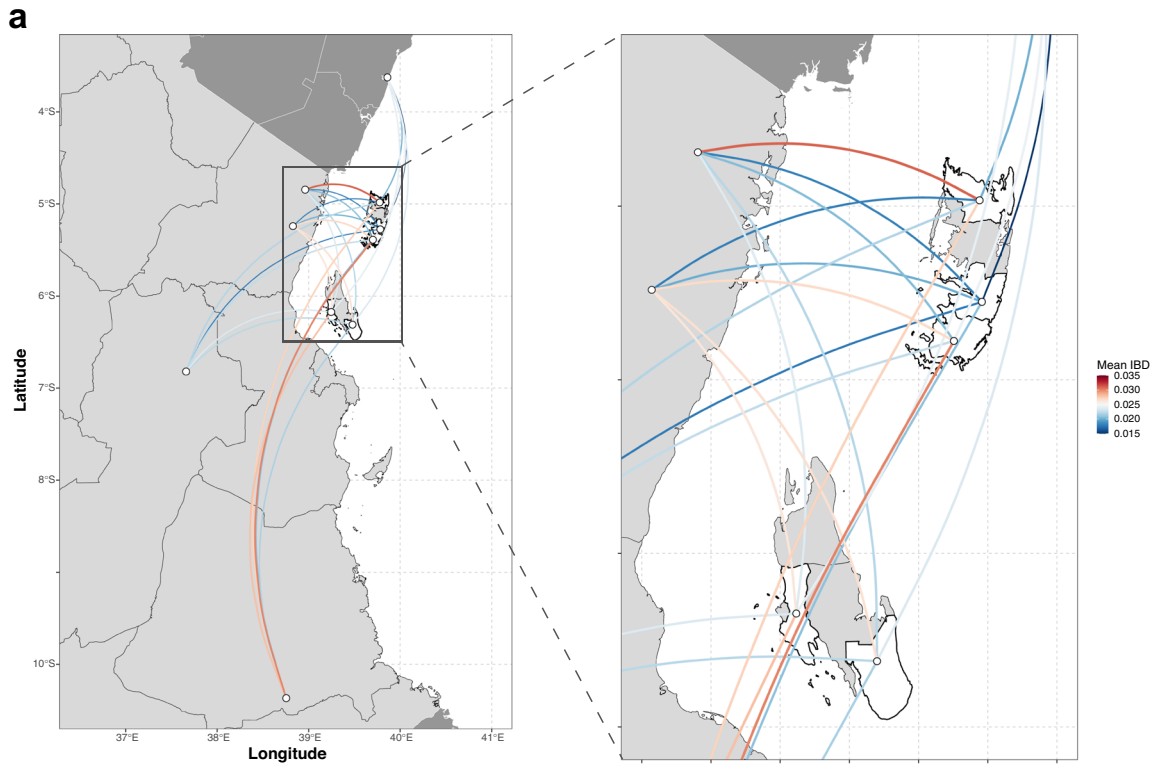

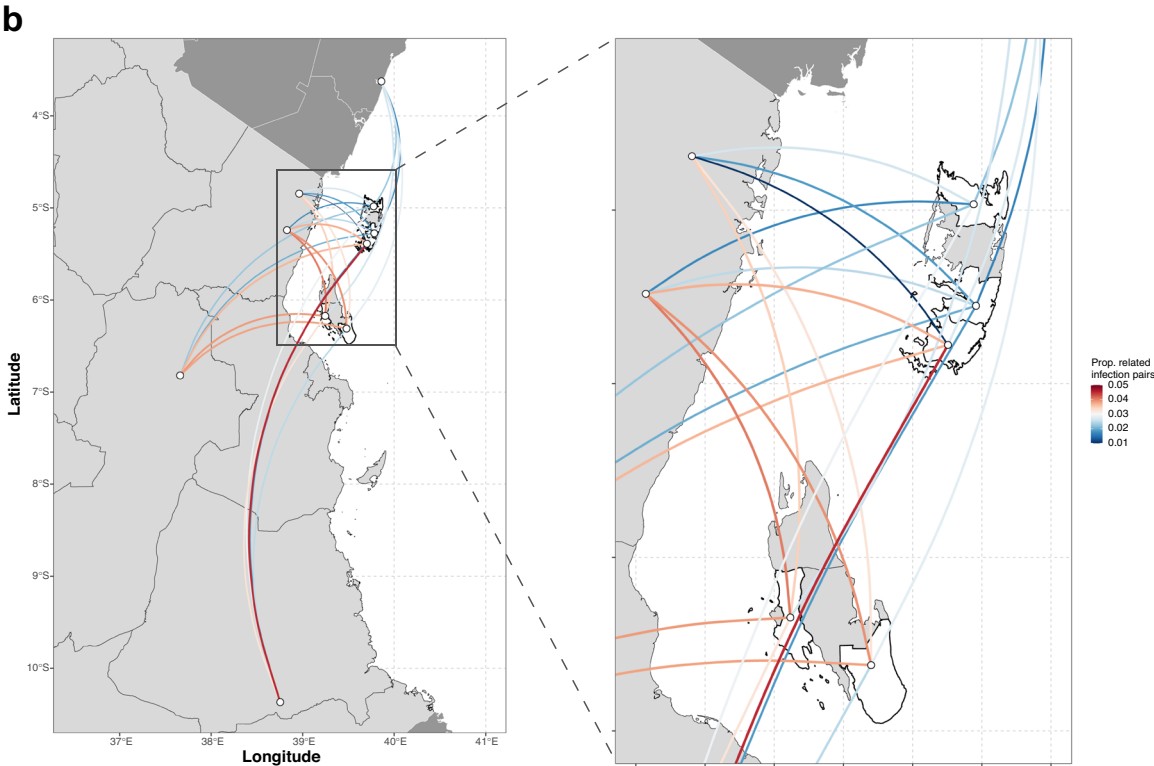

**Fig. 7 | Genetic connectivity between the 5 districts in Zanzibar ($n$ = 290) and the 5 sites on mainland Tanzania ($n$ = 80) and Kenya ($n$ = 35). a** Mean IBD is shown for all pairs. **b** Proportion of related infections is shown for all pairs.

Significance between nodes was assessed using one-sided permutation test (100,000 permutations). Note, no connection was deemed statistically significant ($P$ < 0.05) by permutation testing, likely due to small sample size.

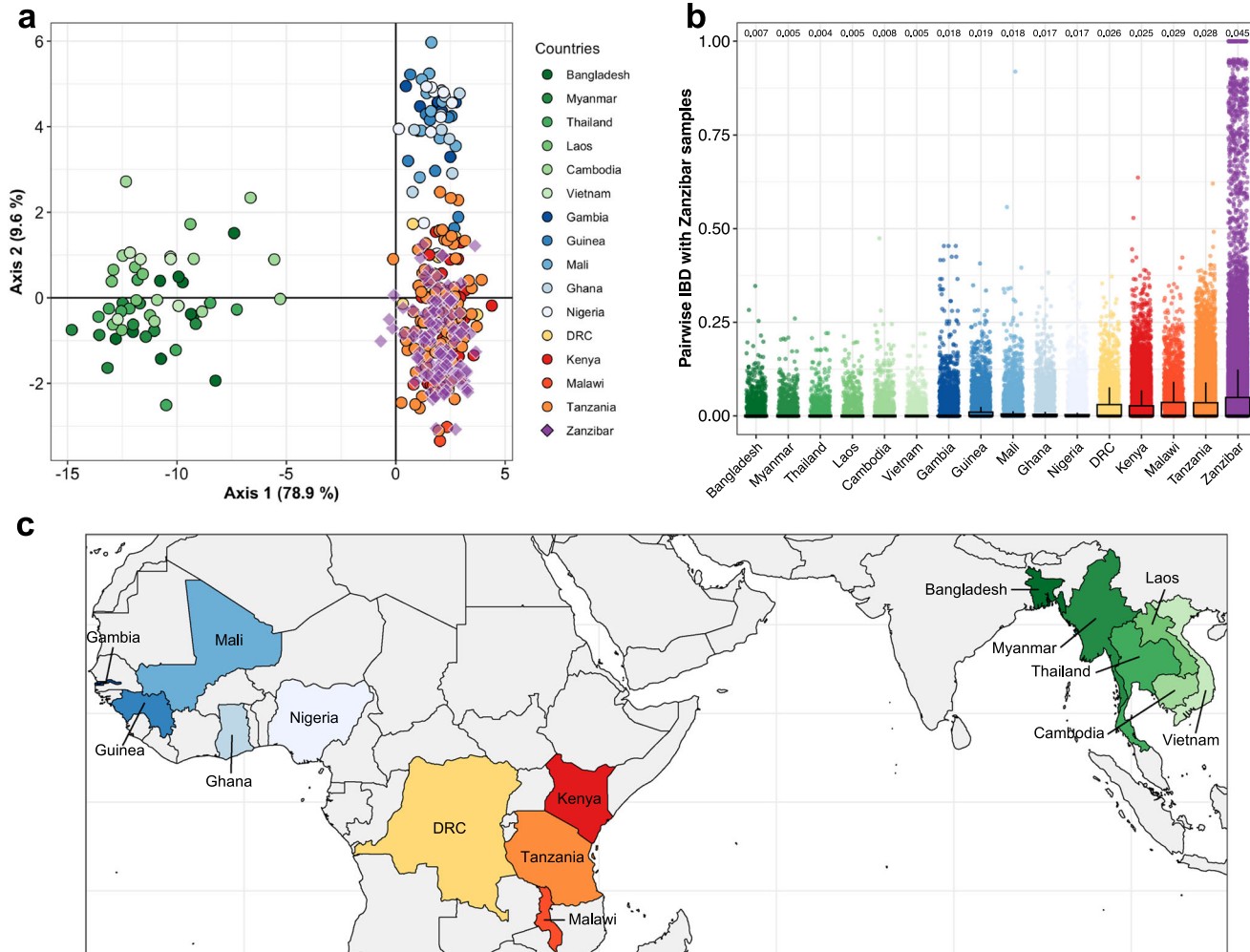

**Fig. 8 | Genetic differentiation and relatedness of *P. falciparum* isolates from Zanzibar and global isolates. a** Genetic differentiation between global *P. falciparum* populations by discriminatory analysis of principal components (DAPC) using 28 microhaplotypes. The first 51 components of a principal component analysis (PCA) are shown. Each point represents a single isolate (*n* = 374); colors indicate country of origin. Only monoclonal samples were used for this analysis. **b** Boxplot showing pairwise IBD between samples from Zanzibar (*n* = 290) and several countries from West Africa, Central Africa, East Africa, and Asia (*n* = 242). Mean IBD is indicated. The box bounds the IQR divided by the median, and Tukey-style whiskers extend to a maximum of 1.5 × IQR beyond the box. **c** Countries from which samples originate are colored in the map, for clarity of the geographic regions under consideration. Democratic Republic of the Congo is abbreviated as DRC.

were more related than between different RACD clusters, indicating that index and additional secondary cases detected during the RACD investigation are often directly linked.

Malaria elimination programs need to target interventions at local scales that are tailored to the drivers of the local transmission situation. More isolated populations with lower parasite diversity and little population movement, such as those identified in Micheweni and Chake Chake districts, would appear amenable to focal control interventions. In comparably isolated foci of local transmission, such efforts are expected to be more sustainable as these areas are less likely to experience regular reintroduction of parasites from other areas. Targeting such areas with a combination of reactive focal MDA and reactive vector control is expected to have a complementary effect and may be more effective than RACD alone, as shown in a trial in Namibia[56]. Indeed, presumptive treatment of household members of malaria cases has been suggested to reduce transmission in Zanzibar, while testing and treating neighbors of malaria cases was shown to only have minimal additional impact on transmission[13]. The results of our study and previous findings from the RADZEC project[2,4,8] have led ZAMEP to actively consider these approaches to

enhance local elimination efforts. ZAMEP has initiated the process of developing standard operating procedures and piloting for reactive focal MDA. On the other hand, despite a signal of predominantly local transmission at smaller scales (within tens of kilometers), we also found that parasite populations in Zanzibar remain highly connected between geographically more distant districts (over tens to hundreds of kilometers), likely driven by human movement. We identified a substantial number of infection pairs at the relatedness level of half-siblings or more (IBD ≥ 0.25), potentially representing imported cases and providing evidence for recent, and most likely ongoing, genetic exchange between Zanzibar and the mainland. This is also supported by the finding that infected individuals detected during RACD most often reported recent travel to a district in mainland Tanzania that exhibits a very high malaria transmission risk[57]. While systems that target malaria infections and transmission at small scale can usefully eliminate local foci, efforts to eliminate malaria in Zanzibar will crucially depend on the reduction of ongoing importation of infections from mainland Tanzania[58]. Targeted efforts in Zanzibar over the past decade maintained incidence on the islands at low levels but a further reduction of transmission in both Zanzibar and

**Table 2 | Frequencies of mutations from drug resistance alleles in the population**

| Gene | Chromosome | Position | Mutation | REF | ALT | Number successfully sequenced | Frequency (%) | | |
|------|-----------|----------|----------|-----|-----|-------------------------------|-----------|--------|-------|
| | | | | | | | Wild-type | Mutant | Mixed |
| *pfdhfr* | chr4 | 748,235 | C50R | T | C | 231 | 100 | 0 | 0 |
| | chr4 | 748,239 | N51I | A | T | 231 | 3.0 | 97.0 | 0 |
| | chr4 | 748,262 | C59R | T | C | 231 | 11.3 | 88.7 | 0 |
| | chr4 | 748,410 | S108N | G | A | 231 | 0 | 100 | 0 |
| | chr4 | 748,410 | S108T | G | C | 231 | 100 | 0 | 0 |
| | chr4 | 748,577 | I164L | A | T | 255 | 100 | 0 | 0 |
| *pfmdr1* | chr5 | 958,145 | N86Y | A | T | 243 | 96.3 | 3.7 | 0 |
| | chr5 | 958,440 | Y184F | A | T | 278 | 33.1 | 45.3 | 21.6 |
| *pfdhps* | chr8 | 549,993 | K540E | A | G | 239 | 16.7 | 82.4 | 0.8 |
| | chr8 | 550,117 | A581G | C | G | 239 | 100 | 0 | 0 |
| | chr8 | 550,212 | A613T | G | A | 239 | 100 | 0 | 0 |
| | chr8 | 550,212 | A613S | G | T | 239 | 100 | 0 | 0 |
| *pfk13* | chr13 | 1,725,370 | I543T | A | G | 265 | 100 | 0 | 0 |
| | chr13 | 1,725,382 | R539T | C | G | 265 | 100 | 0 | 0 |
| | chr13 | 1,725,385 | G538V | G | T | 265 | 100 | 0 | 0 |
| | chr13 | 1,725,388 | N537I | A | T | 265 | 100 | 0 | 0 |
| | chr13 | 1,725,418 | P527H | C | A | 265 | 100 | 0 | 0 |
| | chr13 | 1,725,521 | Y493H | T | C | 265 | 100 | 0 | 0 |
| | chr13 | 1,725,556 | A481V | C | T | 265 | 100 | 0 | 0 |
| *pfmdr2* | chr14 | 1,956,202 | I492V | A | G | 263 | 57.0 | 21.3 | 21.7 |
| | chr14 | 1,956,225 | T484I | C | T | 263 | 100 | 0 | 0 |

Polyclonal and monoclonal samples were included. "Mixed" indicates infections with both the mutant and the wild-type allele.

mainland Tanzania will be necessary to eventually achieve elimination in Zanzibar[58].

Travelers did not cluster separately (Fig. 3) and there was minimal genetic signal of differentiation between samples from mainland Tanzania and Kenya, and Zanzibar. This most likely reflects a source-sink scenario where imported infections from an area of high transmission prevent elimination in areas of lower transmission, as demonstrated in other countries[19,20]. Even though parasite populations between Zanzibar and regions on the mainland of Tanzania remain genetically almost indistinguishable, we find subtle differences between the islands, and some districts. Pairwise IBD sharing revealed that travelers were significantly more related to isolates from mainland Tanzania and Kenya than non-travelers. We were even able to identify differences between travelers vs. non-travelers at the district level on Pemba. However, most of these relationships are far too remote to be attributed to the period covered by the travel history and more likely represent the presence of subtle population structure between Zanzibar and the mainland. Overall, this suggests a high degree of gene flow between Zanzibar and the mainland and likely presents a significant obstacle to reaching elimination on the archipelago.

While denser sampling at sites frequently visited by travelers from Zanzibar might yield further highly related infections, genomic data alone likely is insufficient to differentiate imported infections from those locally transmitted, at such a fine geographical scale with little population genetic differentiation as is the case between Zanzibar and the mainland. We did not collect parasites outside of Zanzibar as part of our study, thus relying on publicly available genomes from the mainland, possibly precluding more fine-scale evaluations of connectivity between Zanzibar and the mainland. Further, sampling of parasites was linked to routine RACD by ZAMEP and hence not spatially uniform across geographic locations. Among the 290 samples successfully genotyped, the proportion of samples from within the same households varies from 25 to 66%, with no clear differences between Unguja and Pemba. While this data is broadly reflective of the

occurrence of cases in the study districts, spatial uncertainty remains in areas with fewer notified index cases and in districts not included in this study.

Our panel was able to distinguish parasites from southeast Asia, east Africa, and west Africa, confirming the validity of the micro-haplotype panel to differentiate samples by continent of origin, though not by country. As expected, the isolates from Zanzibar fell in the east Africa cluster. Estimates of relatedness clearly showed that the Zanzibar isolates are more closely related to samples from east Africa and central Africa than to samples from west Africa. However, they are more highly related within Zanzibar than between any other country.

Drug resistance is a major threat to malaria control. The first-line ACT in Zanzibar has been artesunate/amodiaquine (ASAQ) since 2003, plus single, low-dose primaquine, which to date shows high efficacy in Zanzibar[59]. In Zanzibar, frequencies of known drug-resistance alleles are similar to previous reports in East Africa, including Zanzibar[11,45,59]. Mutations associated with resistance to sulfadoxine-pyrimethamine (SP) in both the *pfdhfr* (N51I, C59R and S108N) and *pfdhps* (K540E) genes were observed at high frequencies, with parasites carrying all four mutations being particularly prevalent. We found high rates of the *pfmdr1* N86 wildtype and 184 F mutant alleles, both of which are associated with decreased sensitivity to lumefantrine[60]. The *pfmdr1* 86Y mutation and Y184 wildtype alleles, conferring resistance to amodiaquine[61], were observed at lower frequencies. The recently reported polymorphism in *pfmdr2* T484I associated with artemisinin partial resistance alleles in Southeast Asia[62,63] was not found. However, the *pfmdr2* I492V mutation was prevalent, which is also suggested to be involved in artemisinin partial resistance[62,63]. Our study did not identify any known *pfk13* artemisinin resistance-conferring mutations, suggesting that parasites in Zanzibar remain sensitive to the current first-line treatment ASAQ. The recent identification of *pfk13* mutants (R561H, C469Y/F, and A675V) in mainland Tanzania[11,23] and neighboring countries Rwanda[26] and Uganda[25], and the high prevalence of mutations in other drug resistance markers found in this study

highlights the need for continued monitoring. Of note, the recently described mutations in the *pfk13* gene found in Rwanda, Uganda, and mainland Tanzania were not covered by our panel.

Our assay performed well on low-density DBS (≥5 parasites/μL) without the need of sWGA pre-amplification prior to the first PCR. The ddPCR-amplicon sequencing approach yielded high read numbers across 35 loci and overall coverage of all markers in samples as low as 10 parasites/μL. More even amplification might be achieved by varying primer concentrations (i.e., primer balancing) to account for amplification rate differences among loci. It was expected that not all qPCR-positive samples could be successfully sequenced as initial parasite screening was done using the multi-copy *var*ATS assay with primers targeting approximately 20 copies per genome[64], and using a direct PCR without prior DNA extraction[8]. This approach is more sensitive than typing of a single-copy marker after DNA extraction[65].

In conclusion, our multiplex AmpSeq panel allowed us to determine MOI, parasite genetic diversity, population structure, genetic relatedness, and to monitor genetic markers associated with drug resistance on the Zanzibar archipelago. Our data presents the most comprehensive genomic assessment of malaria in an island setting closely linked to nearby areas of higher transmission. Our finding of isolated populations on Pemba provides information on fine-scale genetic structure of parasites directly relevant to urgent programmatic needs of the malaria elimination program in Zanzibar.

## Methods

### Study design and participants

The 2022 census reported a population of 1,346,332 in Unguja and 543,441 in Pemba. The islands are highly connected to mainland Tanzania by two airports (international airport in Unguja) and three main seaports (Fig. 2d). In addition to this, there is frequent small boat traffic between the mainland and Zanzibar, often by traditional dhows. Travel to Pemba from mainland Tanzania can be through Unguja, mainly through the Malindi seaport, or directly from Tanga, a northern coastal region in mainland Tanzania, via either Mkoani or Wete seaports. Travelers can also reach Pemba from mainland Tanzania through the airport located at Chake Chake district. As part of its malaria surveillance activities, ZAMEP has implemented RACD since 2012. District Malaria Surveillance Officers (DMSOs) routinely visit households of clinical malaria cases diagnosed at public and private health facilities (index cases) to test all household members by RDT and treat test-positive cases with the first-line anti-malarial artesunate-amodiaquine[4]. The samples used in this study were collected between May 19, 2017, and October 31, 2018, during a rolling cross-sectional survey as part of the ReActive case Detection in Zanzibar: Effectiveness and Cost (RADZEC) study. In the two study years, ZAMEP reported a mean annual parasite incidence (API) per 1000 population of 2.6 (approx. 4000 cases per year; ZAMEP, unpublished data), as compared to an API of 106 reported by the mainland NMCP[66]. The survey covered five districts (Micheweni, Chake Chake, Mkoani, Magharibi, and Kusini) on both Unguja and Pemba, Zanzibar (Fig. 2d). Samples were collected by members of RADZEC study teams who accompanied the DMSOs during their index household follow-ups and extended the RACD procedure to up to nine neighboring households. Detailed travel history within the past 60 days was recorded. The full survey details are described elsewhere[2,8]. DBS samples (*n* = 8933) were screened by direct amplification of *P. falciparum* DNA from 5 × 3 mm DBS punches and quantified using ultra-sensitive *var*ATS qPCR as previously described[8], resulting in a total of 518 positive samples.

Ethical approval for this study was obtained from the Zanzibar Medical Research Ethics Committee (ZAMREC/0001/February/17), the Institutional Review Boards of Tulane University (Study Number: 993573) and of the Ifakara Health Institute (IHI/IRB/No: 003 – 2017), the Ethics Commission of North-western and Central Switzerland (EKNZ Reg-2017-00162), and the University of Notre Dame Institutional Review Board (approval no. 18-12-5029).

### DNA extraction and genotyping of field samples from Zanzibar

Field-collected DBS were stored with desiccant at room temperature until processing. DNA from all 518 *P. falciparum* positive DBS was extracted with the NucleoMag Blood 200 μL kit (Macherey-Nagel) using an optimized protocol for DNA extraction from whole 50 μL DBS, as previously described[65]. Briefly, 240 μL lysis buffer MBL1 and 620 μL PBS were added to each 50 μL DBS and incubated at 94 °C for 30 min. Samples were cooled, 40 μL Proteinase K was added and incubated at 60 °C and 300 rpm for 1 h. The amount of binding buffer MBL2 was adjusted to 800 μL, buffer MBL4 was omitted, and the protocol was followed according to the manufacturer's recommendations. DNA was eluted in 50 μL buffer MBL5 and stored at −20 °C until use.

### Multiplexed ddPCR amplicon sequencing

We developed a novel 35-plex droplet digital PCR (ddPCR) AmpSeq protocol, targeting 28 microhaplotypes with high values for expected heterozygosity ($H_E$) and 7 drug resistance loci distributed across the genome (Supplementary Fig. 18, Supplementary Tables 12 and 13). Primer sequences were designed manually or derived from published protocols with minor modifications to match melting temperature[32] (Supplementary Table 14). All 518 samples were sequenced in duplicate. Amplification of the 35-plex PCR was performed using the BioRad QX200 ddPCR system. A 22 μL reaction mix was prepared containing 4 μL gDNA, 1X BioRad Supermix for Probes (no dUTPs), and 227 nM of each primer (Supplementary Table 15). The reaction mix was then partitioned into 15,000–20,000 microdroplets, followed by a two-step primary multiplex PCR (Supplementary Table 16). Amplification in microdroplets allows for more even amplification of all markers. After amplification, PCR amplicons were recovered from droplets using a chloroform protocol. Briefly, droplets of each sample were homogenized with Tris-EDTA buffer (10 mM Tris, 1 mM EDTA, pH 8.0) and chloroform and then centrifuged at 16,000 rpm for 10 min to separate the phases into an oil phase (bottom) and an aqueous phase (top) containing the amplicons. Samples were then purified using 1.8X AMPure XP beads (Beckman Coulter) and resuspended in 20 μL Buffer EB (Qiagen).

A secondary indexing PCR was performed in a total reaction volume of 50 μL containing 15 μL of purified primary PCR product, 300 nM of each Illumina TruSeq i5/i7 barcode primers (Supplementary Table 17, Supplementary Data 3), 1X KAPA HiFi buffer, 0.3 mM KAPA dNTP mix and 0.5U KAPA HiFi DNA polymerase. Cycling conditions were as follows: initial denaturation 95 °C for 3 min, 10 cycles for samples ≥100 parasites/μL (or 12 cycles for samples <100 parasites/μL) of 20 s denaturation at 98 °C, 15 s annealing at 65 °C and 45 s elongation at 72 °C, and a final elongation of 2 min at 72 °C. Samples were purified with 1X AMPure XP beads (Beckman Coulter) and evaluated using Qubit and Bioanalyzer to assess fragment profile and overall output. Samples were further quantified by qPCR using NEBNext Library Quant Kit for Illumina (NEB) and then pooled equimolarly based on estimated concentrations. The final library pool at a concentration of 1.6 pM, including 25% *PhiX* control v3 spiked in, was sequenced with 150 bp paired-end clusters on a NextSeq 500 instrument (Illumina) according to the manufacturer's recommendations.

The targeted amplicon deep sequencing data was analyzed using the *HaplotypR* package implemented in R, version 0.3 (https://github.com/lerch-a/HaplotypR.git), using filter criteria defined and explained in ref. [29]. Briefly, sequencing reads were demultiplexed by sample and amplicon. Overlapping paired end reads were merged via the *vsearch*

v2.22.1 package[67] and clustered via the *Swarm v2* package[68]. Nucleotide positions of the amplicons were only considered for haplotype calling, if they had a known SNP in *PlasmoDB* (https://plasmodb.org), or if they had a novel SNP that was dominant (>0.5 mismatch rate to the reference sequence) and was supported by at least two samples or replicates. A mismatch rate of >0.5 to the reference sequence indicates a potential novel SNP found on the dominant clone or a clone with a within-host frequency of >0.5 at MOI >1. For single clone infections a mismatch rate of 1 is expected. Chimeric reads, singletons, and reads containing an insertion or deletion were excluded. Haplotype calling from the remaining reads required a within-host haplotype frequency of ≥1% and a minimum coverage of ≥10 reads per haplotype in ≥2 samples or replicates. To further reduce false positive haplotype calls, each haplotype had to be confirmed by both replicates of a sample.

## Protocol validation on mock samples

We generated mock samples from parasite culture lines 3D7 and FCB. Parasite cultures were mixed with uninfected human whole blood to generate a mock *P. falciparum* whole blood mixture of $10^5$ parasites/µL blood. This mixture was then further diluted with uninfected human whole blood to obtain serial dilutions with parasite densities of $10^4$, $10^3$, $10^2$ and 10 parasites/µL blood. Similarly, we also generated mock DBS by spotting 50 µL mock whole blood mixtures on Whatman 3MM filter paper (GE Healthcare Life Sciences) at parasite densities of $10^3$, $10^2$ and 10 parasites/µL blood. DNA was extracted using the NucleoMag Blood 200 µL kit (Macherey-Nagel).

We also generated mixed-strain controls for minority clone detection. 3D7 and FCB strains at similar parasite density ($10^5$ parasites/µL) were mixed at 60:40% (i.e., 3:2 3D7:FCB ratio) and 80:20% (4:1) ratios to mimic polyclonal field isolates. The ratios were serially diluted to $10^3$, $10^2$ and 10 parasites/µL. Furthermore, we included previously described mixed strain controls of 3D7:HB3[29] with a relative abundance of 2% (50:1) and 1% (100:1) for the minority clone (HB3) at a density of $10^3$ parasites/µL.

We compared coverage of markers, fold-difference between the highest number of reads and lowest number of reads per sample, and the proportion of reads lost after trimming of low-quality reads between ddPCR and conventional PCR using *P. falciparum* culture line 3D7 on a range of different parasite densities (500, 50, and 5 parasites/µL).

## Within-host and population level genetic diversity

Within-host diversity was determined using multiplicity of infection (MOI), defined as the highest number of alleles detected by at least two of the 28 microhaplotype loci. Population genetic diversity was estimated from the 28 microhaplotype loci by expected heterozygosity ($H_E$) using the formula, $H_E = \left[\frac{n}{n-1}\right][1 - \sum p_i^2]$, where $n$ is the number of genotyped samples and $p_i$ is the frequency of the $i^{th}$ allele in the population. MOI and $H_E$ were compared by district, island, year, and travel history. We calculated the probability (i.e., $H_E$) that two clones share all 28 haplotypes based on all apparent single-clone infections by concatenating the 28 microhaplotypes into one multi-locus haplotype for each monoclonal sample with data in all 28 microhaplotypes ($n = 28$) and for 27 microhaplotypes ($n = 41$, marker *t01* excluded due to lower coverage). $H_E$ for 28 microhaplotypes combined in all single clone infections was 0.9915, and 0.9921 for 27 microhaplotypes. Thus, the probability that two unrelated clones cannot be distinguished, and such an infection is falsely counted as single clone infection, is less than 1%. Data was visualized using ggplot2.

## Population structure and genetic differentiation

We evaluated geographic clustering by discriminant analysis of principal components (DAPC) using the *adegenet*[69] and *ade4*[70] packages. To avoid overfitting, alpha score optimization was used to determine the appropriate number of PCs to retain for DAPC. We inferred *P. falciparum* population structure (estimated from dominant alleles) using Bayesian clustering package *rmaverick* v 1.1.0[71], under the admixture model with 10,000 burn-in iterations, 2000 sampling iterations and 20 thermodynamic rungs. We estimated genetic differentiation (from dominant alleles) between districts using *Jost's D*[72] and $G''_{ST}$[73], a modified version of Hedrick's $G'_{ST}$ that corrects for a bias that stems from sampling a limited number of populations. Briefly, *Jost's D* and $G''_{ST}$ were calculated using the formulas: $D = \left[\frac{n}{n-1}\right]\left[\frac{H_T - H_S}{1 - H_S}\right]$ and $G''_{ST} = \frac{n(H_T - H_S)}{(nH_T - H_S)(1 - H_S)}$, respectively, where $H_T$ and $H_S$ are the overall and the subpopulation heterozygosity, respectively, and $n$ is the number of sampled populations. The values of *Jost's D* and $G''_{ST}$ range from 0 (no genetic differentiation between populations) to 1 (complete differentiation between populations). As a measure of inbreeding in each population we computed the multilocus linkage disequilibrium (mLD) using LIAN version 3.7, applying a Monte Carlo test with 100,000 permutations[74]. Monoclonal and polyclonal infections (dominant alleles only) with complete haplotypes were included. All analyses were performed using the 28 microhaplotypes only, unless otherwise noted.

## Pairwise genetic relatedness by identity by descent (IBD)

We estimated pairwise genetic relatedness between infections from the 28 microhaplotypes using *Dcifer*, an IBD-based method to infer the degree of shared ancestry between polyclonal infections[42], implemented in R package *dcifer*, version 1.2.0 (https://cran.r-project.org/web/packages/dcifer/index.html). *Dcifer* allows for unphased multi-allelic data such as microhaplotypes, explicitly accounts for MOI and population allele frequencies, and does not require densely distributed or linked markers. Additionally, *Dcifer* provides likelihood-ratio *P* values adjusted for one-sided tests. Briefly, from the samples, naïve MOI estimates and subsequent estimates of population allele frequencies adjusted for MOI were calculated. We used a likelihood-ratio test statistic to test the null hypothesis that two samples are unrelated ($H_O$: IBD = 0) at significance level $\alpha = 0.05$ (with the procedure adjusted for a one-sided test). We have calculated the statistical power and false-positive rate (FPR) of our 28 microhaplotype panel at different relatedness levels using two simulation schemes described elsewhere[42]: (a) no genotyping errors; true MOI and population allele frequencies used as inputs to *dcifer*, and (b) including genotyping error with fixed error model parameters; MOIs and allele frequencies estimated from these data. The FPR was below the nominal significance level $\alpha = 0.05$ (Supplementary Fig. 19). As expected, power was greater for higher values of relatedness and higher MOI led to lower power. For our panel of 28 microhaplotypes, the power to detect siblings ($r = 0.5$) in a pair of infections with MOI of 2 was 0.91, and 0.72 in a pair of infections with MOI of 3; in our study MOI between most infection pairs was rarely above 2–3. We evaluated the mean IBD and the proportion of related samples within and between districts, ward-level administrative units (shehia) and RACD clusters, and performed one-sided permutation test (100,000 permutations) to determine which combinations had higher mean IBD and a higher proportion of related samples than expected by chance. Sample pairs were also categorized into groups based on geographic distance into 10 km bins, with an additional category at distance 0 km to capture within-household comparisons. Mean IBD was calculated for each group. We calculated the odds ratios (OR) for finding significantly related infections between or within, shehia, clusters of households covered during an RACD investigation, and individual households. Comparisons of proportions of highly related infection pairs (IBD ≥ 0.9) within and between different groups were assessed using Pearson's Chi-square test. The *igraph* package in R[75] was used to construct highly related components (referred to as clusters in the results), and to visualize sample pairs at different IBD thresholds to explore them in terms of their spatial and temporal distribution.

## Public sequencing data

We used WGS data of 242 monoclonal samples ($F_{WS} > 0.95$) from the MalariaGEN *Plasmodium falciparum* Community Project[76] to extract microhaplotype data using fasterq-dump (v.2.10.8). Specifically, we used isolates from other sites in Tanzania and Kenya and other regions, including East and West Africa, Southeast Asia, and South Asia (accession numbers in Supplementary Table 18). We extracted microhaplotype sequences using BWA-MEM[77], SAMtools[78], and BEDTools[79] software. Briefly, reads were aligned to the *P. falciparum* 3D7 v3 reference genome assembly using 'bwa mem'. We then used *samtools* library 'sort', 'index' and 'view' functions to extract micro-haplotype regions. The *bcftools* library 'mpileup', 'view' (DP > = 20, MQ > = 30), 'filter' (%QUAL < 99) and 'consensus' functions, and *bed-tools* 'getfasta' were used to extract final microhaplotypes (Supplementary Data 4).

## Analysis of drug resistance loci

We measured the prevalence of drug resistant parasites in Zanzibar, focusing on five markers that have previously been associated with resistance to antimalarial drugs: *pfdhfr* (PF3D7_0417200), *pfdhps* (PF3D7_0810800), *pfmdr1* (PF3D7_0523000), *pfmdr2* (PF3D7_1447900), and *pfk13* (PF3D7_1347700). Mutations in *pfdhfr* and *pfdhps* mediate resistance to the antifolate drugs sulfadoxine and pyrimethamine[80]. The *pfmdr1* N86 wildtype and 184 F mutant alleles are associated with decreased sensitivity to lumefantrine[60], while *pfmdr1* 86Y mutation and Y184 wildtype alleles confer resistance to amodiaquine (AQ)[61]. Polymorphisms at positions T484I and I492V in the *pfmdr2* gene have been suggested to be involved in artemisinin partial resistance in Southeast Asia[62,63]. Mutations in *pfk13*, including R539T and I543T, are strongly associated with artemisinin partial resistance, resulting in delayed parasite clearance[63]. Though associated with AQ resistance[61], *pfcrt* was not typed as no amplification product could be obtained with our assay.

The mutation frequencies in the population were calculated as the number of samples that contained the mutation over the total number of samples successfully genotyped at each locus. Samples that contained the wild type allele and the mutant allele were classified as "mixed". Comparisons of prevalence of mutant alleles across districts were performed using Pearson's Chi-square test. The *UpSetR* package in R[81] was used to visualize different combinations of multidrug-resistant parasites. This analysis was limited to monoclonal samples only with complete allele calls at all loci, because haplotype phasing in complex infection was not possible.

## Reporting summary

Further information on research design is available in the Nature Portfolio Reporting Summary linked to this article.

## Data availability

All raw sequencing data is available at the NCBI SRA (Accession number: "PRJNA954767"). De-identified datasets generated during the current study and used to make all figures are available as supplementary files or tables. This publication uses data from the MalariaGEN *Plasmodium falciparum* Community Project as described in 'Pf7: an open dataset of *Plasmodium falciparum* genome variation in 20,000 worldwide samples', MalariaGEN et al.[76]. Datasets extracted from *PlasmoDB* (https://plasmodb.org) can be accessed using the weblink. All maps presented in this study are generated by the authors; no permissions are required for publication. All analyses were completed in R version 4.1.2.

## Code availability

All analyses were completed in R version 4.1.2 using published R packages, as described in the methods section. This study did not generate any novel code. Code for analyses and raw analysis results are available upon request.

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

## Acknowledgements

We would like to thank everyone who participated in our study, the Shehia leaders in the study districts, all members of the study teams including the DMSOs in the study districts, and the colleagues at ZAMEP for their support. We also would like to thank George Greer from USAID/PMI for his support and contributions at the start of this project. This analysis was supported by RTI International through *Okoa Maisha Dhibiti Malaria* (OMDM) activity (Cooperative agreement number: 72062118CA-00002) in collaboration with the President's Malaria Initiative, the U.S. Agency for International Development and the U.S. Centers for Disease Control and Prevention. The opinions expressed herein are those of the authors and do not necessarily reflect the views of the President's Malaria Initiative, the U.S. Agency for International Development, the U.S. Centers for Disease Control and Prevention, or other employing organizations or sources of funding. A.H. is supported by an Eck Institute for Global Health fellowship.

## Author contributions

A.H. contributed to laboratory analysis, data analysis, generating figures, writing and experimental design. A.L. and I.G. contributed to software design and data analysis. B.S.F., A.H.A., A.A., F.A., M.H.A. and M.A.A. contributed to planning, coordinating, and implementing of field studies to collect samples and clinical data. E.J.R. contributed to study conception and design and interpretation of findings. J.Y., M.W.H., and C.K. contributed to conceiving, planning the study, conducting epidemiologic and bio-specimen data collection, analysis, and interpretation and editing of the final manuscript. All authors reviewed and approved the final version of the manuscript.

## Competing interests

The authors declare no competing interests.
