## [Peer Review File · Nature Communications]

Multiplexed ddPCR-amplicon sequencing reveals isolated *Plasmodium falciparum* populations amenable to local elimination in Zanzibar, TanzaniaREVIEWER COMMENTS

Reviewer #1 (Remarks to the Author):

Holzschuh et al have devised a novel amplicon panel which is able to genotype 35 microhaplotype dispersed across the plasmodium genome. They have validated this panel against mock dilutions of Plasmodium to test sensitivity and have subsequently applied it to a large (n=518) dataset of malaria genotypes derived from dried blood spots.

Patterns of relatedness and diversity indicate frequent travel between Zanzibar and mainland Tanzania and a sink-source relationship between them, while more Northern districts of Pemba show greater isolation from the mainland and may be more amenable to focal interventions.

The paper is well written and the validation appears to have been extensively carried out, showing the utility of their method for identifying COI; their conclusions based on this appear to be robust, and both the results and the tool are likely to be of interest to the wider field, both in academia and public health. However some issues are identified vis-a-vis the significance ascribed to this paper in the context of others in the field, and the interpretations of IBD signals in their panel, which should be rectified prior to publication.

The significance of the findings and the applications of the method are overstated in several places. The claim that "most studies utilizing this method have only targeted one or a few loci" (L107) is manifestly untrue. Tessema et al (JID 2022) have made a far larger panel for exactly this kind of inference which also utilises microhaplotypes yet types ~100 loci allowing more accurate inference of relatedness and diversity and including some of the same resistance loci. I find it hard to justify this assertion. A direct comparison of these two panels would be welcome in this paper. Similarly, the claim that this is the first examination of Pf in an island close to the mainland with higher transmission (L128) is arguable, but it is a fine distinction at best. Both Daniels 2020 and Mze 2020 (both MalJ) have examined island populations, the latter next to a region of high transmission, while numerous papers have used genomics to examine importation / cross-border transmission

IBD signals have been calculated using previously tested methods which are generally robust, however I have some concerns about the relatively small number of markers used here, and the high proportion of them concerned with drug resistance or cytoadherence.

I see no distinction made within the methods between drug resistance loci and microhaplotype markers, though selection acting on the drug resistance loci will inevitably skew IBD inference (loci involved in cytoadherence are also a risk here). Where inbreeding and LD are used to infer epidemiological outcomes in particular (L267 / L255 / L433), were resistance loci removed? Similarly CNVs can have a dramatic effect on IBD and diversity metrics and would be expected to be enriched in high He regions; the extremely high proportions of 'mixed' genotypes for MDR1/2 are also likely to relate to known CNVs at these loci. What was done to control for these features both within drug resistance loci and within the microhaplotype loci?

Minor queries / comments follow:

L566 - Power to detect relatedness between sites has been effectively tested in this paper, however little appears to be done to quantify power *within* sites? How does panel IBD compare to known relatedness in different contexts wrt lower diversity or cardinality of markers?

L244 - how divergent are 3D7 and HB3 as measured by WGS and AmpSeq? How many of the amplicons remained informative in this comparison?

L248 - MOI is widely used, but may be simplistic in an island setting with expected higher degrees of inbreeding. However this could be estimated from the data at hand: If all monoclonal haplotypes were mixed at the proportions found, what would be the FDR of monoclonality, or the degree of undercounting wrt MoI?

L437 - 'significant' relatedness - Dcifer Pvalue < 0.05?

L541 - Fig 7, the dominance of links to Zanzibar makes this difficult to interpret - could Zanzibar be shown inset at larger scale?

Reviewer #2 (Remarks to the Author):

This article on the use of ddPCR AmpSeq to elucidate malaria transmission patterns in Zanzibar was a joy to read, a well written and presented case of genetic epidemiology of malaria at fine geographic scales.

The authors achieve a good rate of genotyping success considering the very low parasitaemias analyzed and the absence of sWGA pre-amplification from the method.

Given that this work is partially presented as a methods paper, a few extra words on the pros/cons of ddPCR vs other AmpSeq approaches may be interesting to readers.

For example I am curious if there is data available to show how sensitivity and evenness of marker coverage differs between conventional PCR and ddPCR implementations of this method, and if most conventional PCR multiplexes can easily be converted into the ddPCR format if valuable to do so.

The article describes various inferences achieved from the 28 microhaplotypes analyzed, especially based on relatedness (IBD) calculation applied to polyclonal samples. For example, subtle variation in mean and fraction high IBD levels indicate malaria spread within shehias and households, malaria spread across the dry season, an admixture zone on Unguja island, and slightly higher island-mainland infection relatedness in recently traveled vs. non-travelling patients. Some inferences, e.g., patterns of connectivity between mainland and island populations appear more speculative in my appreciation. The discussion does well to summarize main inferences (e.g., ongoing parasite dispersal from the mainland to the islands as an obstacle to malaria elimination), although it was less clear how MOI and heterozygosity/diversity metrics fit the narrative and which/how patterns exactly are 'actionable' in this study.

My remaining comments are mainly cosmetic:

Line 45: The number 518 is a bit misleading here given that it was 290 whose sequencing results sufficed for pop gen analysis.

Line 46: Perhaps 'islands of Zanzibar' instead of just 'islands'.

Line 54: Is 'related' here different than 'highly related' in line 51?

Line 72: Within this intro of malaria in Zanzibar, is it possible to add case numbers/year on mainland and islands?

Line 75: I think as worded here the text is asserting that in lower-transmission intensity settings most malaria infections are asymptomatic, and most asymptomatic infections have parasite densities below the limit of detection of RDTs. Is this true as worded?

Line 99: 'Unmatched' resolution wouldn't be true unless you are referring only to non-WGS methods.

Line 102: Can't SNPs be multi-allelic? Are you refereeing to number of polymorphic nucleotide positions within the analyzed marker, or number of alleles within the analyzed marker? (the former I'm not sure actually fits the meaning of 'multiallelic')

Line 151: It would be interesting to add these seaports as points on at least one plot. Or mention their names so readers can follow up. Also would be interesting to know if travel to Pemba is generally via Unguja or also commonly directly from mainland.

Line 223: What is >0.5 mismatch rate? Would >0.5 & <1 refer to a heterozygous call of a novel SNP?

Line 243: Why did you not create the 1-2% strain ratios for 3D7/FCB?

Line 248: Why at least 2 loci?

Line 338: Does 'detection of alleles' mean the same thing as % loci amplified (line 335)?

Line 339: Does 'down to' mean 'at'?

Line 346: I suggest to add the numbers for 'still high', else it sounds evasive I think.

Line 349: Did lowering the detection limit to 0.1% lower precision of reference strain genotypes?

Line 355: Is there a histogram etc. in supplement to show parasite densities of the DBS sample set?

Line 385: What is inferred from the strong correlation between observed het. and global exp. het.? Could the authors add a comment on this here or in the discussion?

Fig. 1B: Could you add DBS sample sizes for each bin or otherwise state this clearly somewhere in beginning of DBS results?

Fig. 2A: Possibly add mainland to map for context?

Fig. 2B: From the methods I understood that MOI was classified as an integer? Why are the sample points in the boxplot predominantly decimals while the medians appear to be integers 1 or 2? Sorry if I've missed something here.

Fig 2B/2D: Why do sample size appear different between these 2 graphs?

Line 422: Perhaps 'local transmission' could be reworded to something like 'locally restricted transmission'.

Fig 3. Very nice plot. You could possibly label Pemba/Unguja since these are referred to in text. Possibly could add a 2-plot version split by travel/no travel to supplement? It's hard to see here if there is any subclustering among non-travellers around the 0,0 origin.

Line 456: Perhaps clarify "among non-travelers vs. travelers" = "among non-travelers vs. among travelers".

Line 531: Hard to see how these correspond well. To me it looks like the western region of Unguja near port of entry has higher IBD to mainland, as well as southern Pemba, though not the next adjacent district in Pemba which has low IBD to mainland but similar travel affinity based on the questionnaires.

Fig. 7: What are the sample size for each geographical node in this analysis? How much does # of pairwise comparisons differ among the edges and may this distort inference? E.g., do you get the same results when balancing sampling randomly or applying some sort of bootstrapping approach?

Was WGS data from localities on the mainland closer to the islands (e.g., coastal Pwani) not applied in IBD analyses because they were not available from WGS or because travel there is not common based on the questionnaires? Or other reasons that area is less relevant...?

Fig. 9B: Again couldn't hurt to add island labels.

Line 625: Aside from spatially correlated dispersal, do any metrics of this study allow inference of transmission intensity differences? E.g., between Pemba and Unguja?

Reviewer #3 (Remarks to the Author):

Holzschuh and colleagues describe a study of genetic variation in *Plasmodium falciparum* malaria parasites in Zanzibar. This study makes two important contributions. Firstly, the study describes the use of a novel amplicon panel to sequence and assess genotypes at 28 microhaplotype and 7 drug resistance loci. This panel could be useful to others in the community. Secondly, the study highlights a number of important results, which include a) patterns of genetic variation on Ugunja island are largely consistent with parasites being imported from mainland Africa, b) patterns of genetic variation on Pemba island show a more isolated population suggesting greater local malaria transmission, c) individuals living closer together are more likely to carry highly related malaria parasites, and d) although many drug resistance markers are present, parasites in Zanzibar should largely be sensitive to current first-line antimalarial treatments.

In contrast to many studies of genetic variation in malaria parasites, this paper offers an intriguing example of how the results might be translated to actionable intelligence - the greater evidence for local transmission, rather than importation, on the island of Pemba suggest this might be a good candidate for a campaign of mass drug administration (MDA), whereas such a campaign on Unguja might be less successful.

In general the paper is well written, and the conclusions are supported by the data. The methodology is largely sound and the methods section provides enough detail for the work to be reproduced.

Major comments

1) For me the most interesting aspect of this paper is that there appears to be evidence of local transmission in north Pemba, but this is less evident in Mkoani or on Unguja (as evidenced by Figure 3 and Figure S8A/B). An alternative explanation for these results, however, could be that the sampling framework was different in north Pemba, for example many more samples from the same households (RACD samples) were taken in Micheweni and Chake Chake than in other districts. Indeed, on lines 694-695 it is stated that "sampling of parasites was linked to routine RACD by ZAMEP and hence not spatially uniform across geographic locations", but details are not given. The authors should provide a breakdown of samples from the same vs different households by district and year so this alternative explanation could be ruled out.

2) I am somewhat sceptical that reliable measures of IBD can really be determined with just 28 genetic markers. For example, the inset of Figure 4A shows a perhaps surprising number of sample pairs with mean IBD between 0.75 and 0.9. This would suggest these are not clonal pairs but also that they are more highly related than, for example, sibling pairs. This could be evidence of multi-generational inbreeding but might also be an artefact of attempting to estimate genome-wide IBD with a small number of markers. Figure 4 of the Dcifer paper (reference 42) suggests confidence intervals for relatedness measures when using < 50 loci are likely to be very high. Moreover, even in a study of a highly-related South American populations using whole genome data (<https://pubmed.ncbi.nlm.nih.gov/36542676/>), very few sample pairs have IBD between 0.75 and 0.9. The authors should comment on the somewhat surprising nature of the IBD estimates in their study, for example by stating whether they think they have evidence of multi-generational inbreeding or whether they think this is an artefact

3) Also on the topic of IBD, I am sceptical that the apparently highly related samples between Zanzibar and mainland Africa shown in Figures 6, 7 and 8 are truly highly-related and not apparently highly related by chance due to the low number of genetic markers used. One thing that would give the reader more confidence would be showing that there are no highly related sample pairs between samples from Zanzibar and those from elsewhere in the world, e.g. from Asia. For example, Figure 9B could also include pairwise IBD with samples from Asian countries.

Minor comments

- 4) The gene CRT was not included in panel, despite AS-AQ being used for treatment (AQ has been associated with K76T in some studies). I think many readers might be surprised to see the omission of this gene from the panel, so a short description of why this was not included might be helpful.
- 5) The coverage profile of different amplicons (Figure 1C) shows large variation. As an example the average number of reads for c27 appears to be over 100x greater than that for t01. This might not be immediately obvious to many readers because the y-axis of this figure is on a log-scale. It is recommended that either the figure be recreated with a non-log scale, or that at least it is clearly highlighted that a log scale is used which might make the variation between amplicons appear to be lower than is actually the case.
- 6) The large variation in coverage between amplicons seen in Figure 1C should be highlighted in the main text, together with any suggestions as to how more even coverage might be achieved (e.g. through primer balancing). In particular, I don't think the statements "high level of uniformity in the average number of reads per target" on lines 366-367 and "relatively even amplification" on line 730 can be supported and should be removed.
- 7) In Table S1, the median and IQR of reads per locus are given, but I think the range might also be useful information here, or at least 5% and 95% quantiles.
- 8) In Figures S2B and S2D, coverage needs to be defined. Is the coverage at 1X, 10X or something else?
- 9) In Figure S2B it appears that some points appear to be slightly greater than 100%. Is that true, and if so, how can coverage be greater than 100%?
- 10) MOI is defined as "the highest number of alleles detected by at least two of the 28 microhaplotype loci". As such MOI for each sample should be an integer, but in Figure 2B the values are not integers. This requires an explanation.
- 11) Some points on Figure 2B have MOI < 1. Is this a mistake? If not, what is the biological meaning of a sample with MOI less than 1?
- 12) For Figure 2C, the authors might want to consider showing estimates of heterozygosity from a set of east African or Tanzanian samples, as presumably this should have even higher correlation with observed heterozygosity than a global set of samples?
- 13) I presume the statement "Samples in Micheweni share a high proportion between transmission seasons" on line 453 is based on the Micheweni 2017 vs Micheweni 2018 cells in Figure S8C/D. It might be helpful to the reader to refer to that figure here, and maybe put the mean IBD (0.101) and/or Proportion of related infection pairs (0.107) results in the main text.
- 14) The statement "This was still true when we excluded all sample pairs where both individuals reported recent travel" on lines 484-485 states something that presumably would be expected; samples from patients that have travelled are presumably less likely to be related and therefore removing these would mean the samples that remain are expected to more highly related. An arguably more interesting analysis here would be to look at the levels of relatedness only among pairs of samples where one (or both) have travelled. If these showed significantly lower relatedness, that would presumably be of interest? Perhaps such an analysis could be added to Table 1?
- 15) On lines 190 and 256 it is stated that "All samples were sequenced in duplicate", and it is also stated that "A total of 290 (56%) DBS samples with data 356 in ≥ 10 loci for both replicates and a minimum of 10 reads per marker were included in further analyses", but I could see no details of how final results for each sample were arrived at. E.g. was there always perfect concordance between duplicate pairs and if not how were discrepancies resolved? This should be explained in the Methods section.
- 16) On line 318, it might be helpful to explain why mutations R539T and I543T have been singled out, rather than, for example, the more common C580Y mutation or mutations seen recently in

Africa such as R561H, A675V or C469Y

17) On lines 320-321, it is unclear whether the phrase "samples genotyped" at the end of the sentence includes or excludes samples for which genotypes could not be determined (i.e. which have missing genotypes) which is an important distinction. This should be clarified.

18) On line 548 "Three pairs from two individuals with IBD >0.5" are mentioned. Does this refer to the two red lines on Figure 7C? If so, it might be helpful to the reader to state this explicitly. Also, should this read "Two pairs from three individuals"?

19) On line 567-568 it is stated that "Zanzibar samples are more closely related to samples from east Africa and central Africa than to samples from west Africa". Which countries are considered east and central here? And is this statement based on formal analysis or inferred from Figure 8B? My reading of this figure is that IBD measures look somewhat higher from Kenya and Tanzania (which presumably are considered east Africa), but that IBD measures from countries that might be considered central such as Malawi or DRC don't look particularly higher than IBD measures from west Africa. Perhaps the phrase should be changed to "Zanzibar samples are more closely related to samples from east Africa than to samples from elsewhere in Africa" or similar?

20) On lines 597-600 it would be helpful to quantify prevalence of triple mutant and N86-184F haplotypes (% and n).

21) On lines 651-653 it is stated that "Targeting such areas with a combination of reactive focal MDA and reactive vector control is expected to have a complementary effect", but it is not clear whether such approaches are actively being considered in Zanzibar. If possible, I think it would be useful to expand on this, and also give an indication of whether the results presented here are actively being considered for future control and elimination efforts, e.g. by ZAMEP. I feel the manuscript might be of greater interest to many readers if it was clear that this analysis had driven a change in policy.

22) On lines 660-663 it is stated that "We identified a substantial number of infection pairs at the relatedness level of half-siblings or more (IBD ≥ 0.25), likely representing imported cases and providing evidence for recent, and most likely ongoing, genetic exchange between Zanzibar and the mainland". If IBD measures ≥ 0.25 determined in this study are truly evidence for recent genetic exchange between Zanzibar and the mainland, Figure 8B would suggest there has been recent exchange from across Africa. The authors should comment on whether they think this truly is the case.

23) On line 720, the claim that lack of pfk13 mutations confirm that parasites remain sensitive to ASAQ is rather strong. Perhaps the word "confirming" could be changed to "suggesting" or similar?

24) It would have been fascinating to know whether some of the results presented here would have held if whole genome sequencing (WGS) had been used, e.g. would IBD values between 0.75 and 0.9 still be seen, and would IBD ≥ 0.25 still be seen with each African country. Have the authors considered WGS, either themselves or in collaboration with others such as the MalariaGEN network? It might be useful to comment in the Discussion on what limitations the study has from having only 35 markers, and to what extent these might be addressed by WGS.

25) There are supplementary tables named S1-S5 in both Supplementary Results and Supplementary Methods which is confusing. Could these two supplementary documents be combined and tables renumbered accordingly?

26) It might be useful to comment on why a different bioinformatics pipeline was used for the amplicon data (HaplotypR) vs the analysis of WGS data (bwa/samtools)

Richard Pearson, Wellcome Sanger Institute

11 April 2023

Point-by-point responses to reviewer comments

Holzschuh et al., “Highly multiplexed ddPCR-amplicon sequencing reveals strong *Plasmodium falciparum* population structure and isolated populations amenable to local elimination efforts in Zanzibar”

We thank the editors and reviewers for their helpful comments and suggestions for improvement. Original reviewer comments are numbered and in italics, and our responses are indented and in normal font. Additionally, changes to the text are indicated throughout this document and the manuscript in red text. We believe that the changes made in response to these suggestions have led to a substantially improved manuscript.

Reviewer #1:

1) The significance of the findings and the applications of the method are overstated in several places. The claim that “most studies utilizing this method have only targeted one or a few loci” (L107) is manifestly untrue. Tessema et al (JID 2022) have made a far larger panel for exactly this kind of inference which also utilises microhaplotypes yet types ~100 loci allowing more accurate inference of relatedness and diversity and including some of the same resistance loci. I find it hard to justify this assertion. A direct comparison of these two panels would be welcome in this paper.

RESPONSE: We thank the reviewer for this comment. We acknowledge the importance of the work by Tessema et al., which was cited in our original submission. We have rewritten the paragraph and emphasized this as follows (Page 5, lines 102-116):

“Novel amplicon sequencing-based genotyping methods allow typing of parasites **at high with unmatched** resolution and sensitivity and provide a highly detailed picture of the parasite population structure^{27,28}. Targeted amplicon sequencing (AmpSeq) of the most informative regions in the genome allows for deep and consistent sequence coverage²⁹. **AmpSeq offers very high detectability of minority clones as low as 1% frequency in polyclonal infections²⁹, which are frequent even in pre-elimination settings^{20,30}.** By targeting short, highly diverse **microhaplotype loci that contain multiple SNPs and exhibit** multiallelic loci (~~microhaplotypes~~) rather than biallelic diversity loci (e.g. SNPs), discriminatory power and ~~relatedness inference can be increased^{31–33}~~. **Amplicon sequencing offers very high detectability of minority clones as low as 1% frequency in polyclonal infections²⁹ which are frequent even in pre-elimination settings^{20,34}.** To date, many ~~most~~ studies utilizing this method have only targeted one or a few **genomic** loci to gain information on diversity of infections, drug resistance, or selection^{29,34–36}. **Only recently, there have been efforts to extend these methods to large multiplexed AmpSeq panels covering numerous genetically diverse loci (e.g., microhaplotypes) and markers of drug-resistance, as pioneered by**

Tessema et al.³² and others^{15,33}. Increasing the number of **diverse** loci can provide a higher resolution comparison of infections at the population level as well as pairwise relatedness inference at the individual level^{15,32,33}.”

2) Similarly, the claim that this is the first examination of Pf in an island close to the mainland with higher transmission (L128) is arguable, but it is a fine distinction at best. Both Daniels 2020 and Mze 2020 (both MalJ) have examined island populations, the latter next to a region of high transmission, while numerous papers have used genomics to examine importation / cross-border transmission

RESPONSE: We thank the reviewer for highlighting these works. Of note, in the manuscript we did not claim this was the first study, but rather “Few studies have examined the population structure of *P. falciparum* in island settings potentially closely linked to mainland areas with higher transmission.” We have added the two studies to our introduction and referenced them as follows (Page 6, line 134-136):

“~~Few~~ Previous studies **that** have examined the population structure of *P. falciparum* in island settings potentially closely linked to mainland areas with higher transmission **analyzed a limited number of samples and typed only few genomic markers**^{43, 44}.”

We still believe our work differs substantially from these studies. Both studies used a small number of makers (21-SNP barcode in Daniels et al., 2020; msp1/2 and 24-SNP barcode on a subset of samples in Papa Mze et al., 2020), and they did not specifically address the question of importation from nearby sites of higher transmission.

3) IBD signals have been calculated using previously tested methods which are generally robust, however I have some concerns about the relatively small number of markers used here, and the high proportion of them concerned with drug resistance or cytoadherence.

I see no distinction made within the methods between drug resistance loci and microhaplotype markers, though selection acting on the drug resistance loci will inevitably skew IBD inference (loci involved in cytoadherence are also a risk here). Where inbreeding and LD are used to infer epidemiological outcomes in particular (L267 / L255 / L433), were resistance loci removed?

Similarly CNVs can have a dramatic effect on IBD and diversity metrics and would be expected to be enriched in high H_e regions; the extremely high proportions of ‘mixed’ genotypes for MDR1/2 are also likely to relate to known CNVs at these loci. What was done to control for these features both within drug resistance loci and within the microhaplotype loci?

RESPONSE: In all population genetics analyses as well as relatedness inferences, we have excluded the 7 drug resistance loci and only used the 28 microhaplotypes. Since we have not included any drug resistance loci in IBD inference, they cannot

have skewed any inference. Only when we performed DAPC to compare the parasite population from Zanzibar to other populations from Africa and Asia, we have done this with and without the drug resistance loci (see Supplementary Figure 18). We have clarified this as follows:

(Page 13, lines 301-302)

“All analyses were performed using the 28 microhaplotypes only, unless otherwise noted.”

(Page 13, lines 305-306)

“We estimated pairwise genetic relatedness between infections from the 28 microhaplotypes using *Dcifer*, an IBD-based method to infer the degree of shared ancestry between polyclonal infections⁴², implemented in R package *dcifer*, version 1.2.0 (<https://cran.r-project.org/web/packages/dcifer/index.html>).”

We would also like to refer to the power simulations that we have done (see comment #41) noting that the power to detect sibling-level relatedness ($r = 0.5$) and even half-sibling level ($r = 0.25$) is high. The statistical inference provided by *Dcifer* takes the number of markers into account (e.g. when p-values for hypothesis testing are calculated and significantly related pairs are determined), and the error rate control (false positive rate below the nominal significance level $\alpha=0.05$) is good as demonstrated in the *Dcifer* paper (Gerlovina et al., 2022) and also evidenced by our simulations for this paper (see response to comment #41 for more details).

Minor queries / comments follow:

4) L566 - Power to detect relatedness between sites has been effectively tested in this paper, however little appears to be done to quantify power *within* sites? How does panel IBD compare to known relatedness in different contexts wrt lower diversity or cardinality of markers?

RESPONSE: As our focus is on Zanzibar, we kept the discussion of global patterns short. However, we have calculated within-site mean pairwise IBD and the proportion of significantly related infections (see Table below). Global scale studies on *P. falciparum* genetic diversity show that high levels are generally predominant in African populations and intermediate/low in Southeast Asian populations (MalariaGEN et al., 2021; Amambua-Ngwa, A. et al., 2019; Manske, M. et al., 2012). We therefore would expect both, between relatedness and within relatedness, of different countries in high transmission areas to be lower due to frequent recombination. In contrast, in areas with lower malaria transmission, such as Southeast Asia, where *P. falciparum* exhibits lower genetic diversity, likely due to the smaller population size of the parasite in these regions, as well as reduced opportunities for recombination due to lower transmission rates. We would expect within relatedness to be high, but between relatedness to be low, due to geographical barriers and reduced gene flow (genetically isolated from each other). Linkage disequilibrium (LD) differs substantially

between parasites from Asia and Africa, where LD is extended for extremely short physical distances in African parasites but substantially longer in Asian parasites (Volkman, S. *et al.*, 2007, Shetty *et al.*, 2019, MalariaGEN *et al.*, 2021). As expected, we see clear differences in within-site relatedness between Africa and southeast Asia. Mean IBD and the proportion of related infections is substantially higher in southeast Asia and highly significant for all sites. We added the following sentence (Page 29-30, lines 646-649):

“Previous studies on *P. falciparum* genetic diversity on global scale showed that low IBD levels within sites are predominant in African populations and intermediate/high levels in Southeast Asian populations^{68–71}. The same pattern was observed by our panel of markers (Supplementary Table 17).”

Pairs within sites	Mean IBD	P value	Proportion significantly related infections	P value
Southeast Asia				
Bangladesh	0.1762	0.00002	0.7200	<0.00001
Myanmar	0.2377	<0.00001	0.8600	<0.00001
Thailand	0.2913	<0.00001	0.8200	<0.00001
Laos	0.2194	<0.00001	0.7200	<0.00001
Cambodia	0.1637	0.00006	0.4691	<0.00001
Vietnam	0.2948	<0.00001	0.5432	<0.00001
West Africa				
Gambia	0.1452	0.00011	0.2600	0.00011
Guinea	0.0425	0.13757	0.0494	0.45338
Mali	0.0571	0.02930	0.1600	0.00287
Ghana	0.0914	0.00249	0.2000	0.00074
Nigeria	0.0464	0.08012	0.1400	0.00759
Central Africa				
DRC	0.0302	0.38064	0.0400	0.57333
East Africa				
Kenya	0.0283	0.50834	0.0571	0.07680
Malawi	0.0379	0.18783	0.0400	0.57536
Tanzania	0.0262	0.89018	0.0434	0.43101

5) L244 - how divergent are 3D7 and HB3 as measured by WGS and AmpSeq? How many of the amplicons remained informative in this comparison?

RESPONSE: Supplementary Table 8 shows that 3D7 and HB3 are dimorphic at 21/28 of microhaplotype loci, hence 21 loci remained informative. Among those 21 loci, 98 SNPs differ between the two strains. WGS data is publicly available from PlasmoDB and confirms this number. The 3D7 isolate originated from Africa, while

HB3 is from Central America (Honduras), thus those isolates are expected to be significantly divergent.

6) L248 - *MOI is widely used, but may be simplistic in an island setting with expected higher degrees of inbreeding. However this could be estimated from the data at hand: If all monoclonal haplotypes were mixed at the proportions found, what would be the FDR of monoclonality, or the degree of undercounting wrt Mol?*

RESPONSE: We calculated the probability that two clones share all 28 haplotypes based on all apparent single-clone infections by concatenating the 28 microhaplotypes into one multilocus haplotype for each monoclonal sample with data in all 28 microhaplotypes (n=28). We did the same but excluded marker *t01* to increase sample size (n=41), which had lower coverage than all other marker. The probability that two clones are different (by multilocus haplotypes) is 0.9915 for samples with all 28 microhaplotypes and 0.9921 for samples with 27 microhaplotypes (excluding *t01*). We added this info (Page 12, line 276-283):

“We calculated the probability (i.e., H_E) that two clones share all 28 haplotypes based on all apparent single-clone infections by concatenating the 28 microhaplotypes into one multi-locus haplotype for each monoclonal sample with data in all 28 microhaplotypes (n=28) and for 27 microhaplotypes (n=41, marker *t01* excluded due to lower coverage). H_E for 28 microhaplotypes combined in all single clone infections was 0.9915, and 0.9921 for 27 microhaplotypes. Thus, the probability that two unrelated clones cannot be distinguished, and such an infection is falsely counted as single clone infection, is less than 1%.”

7) L437 - *‘significant’ relatedness - Dcifer Pvalue < 0.05?*

RESPONSE: Yes, the reviewer is correct that significantly related pairs are all pairs with a *P* value < 0.05 as estimated by *dcifer*. We have clarified (Page 21, line 500-501):

“Overall, we found 4.1% (1,703) significantly related pairs (4.1%; 1,703/41,905 *P* < 0.05 by *Dcifer*).”

8) L541 - *Fig 7, the dominance of links to Zanzibar makes this difficult to interpret - could Zanzibar be shown inset at larger scale?*

RESPONSE: We agree and have changed Figure 7 so that Zanzibar is shown inset at larger scale. We also decided to move Figure 7C to the supplement (now Supplementary Figure 17), so that Figure 7A and B can be shown at larger sizes.

Reviewer #2:

9) Given that this work is partially presented as a methods paper, a few extra words on the pros/cons of ddPCR vs other AmpSeq approaches may be interesting to readers. For example I am curious if there is data available to show how sensitivity and evenness of marker coverage differs between conventional PCR and ddPCR implementations of this method, and if most conventional PCR multiplexes can easily be converted into the ddPCR format if valuable to do so.

RESPONSE: We have added a comparison of amplification by ddPCR and conventional PCR in **Supplementary Table 10**. To this aim, we compared the number of markers with data obtained using protocols including and excluding droplets, and evenness of read counts. The exploratory comparison was done using two replicates of 3D7 parasite isolates at three different parasite densities (500, 50, and 5 parasites/ μ L). Both protocols successfully amplified samples at all densities. The evenness, i.e., the fold-difference between the highest number of reads and lowest number of reads per sample is better for the ddPCR assay, ranging from 23X - 220X for ddPCR and 429X - 1,035X for conventional PCR. Therefore, using ddPCR resulted in more even coverage of markers, increasing the chances of detecting minority alleles. In addition to this, the proportion of reads that are lost after trimming of low-quality reads (e.g., amplification artifacts) seems to be substantially higher when using conventional PCR, especially at lower densities (at 5 parasites/ μ L: 4.1-5.1% for ddPCR and 10.2-11.0% for conventional PCR). We have added the following to the Methods and Results sections:

(Page 12, line 265-268)

“We compared coverage of markers, fold-difference between the highest number of reads and lowest number of reads per sample, and the proportion of reads lost after trimming of low-quality reads between ddPCR and conventional PCR using *P. falciparum* culture line 3D7 on a range of different parasite densities (500, 50, and 5 parasites/ μ L).”

(Page 16-17, line 398-404)

“More even amplification of markers using the droplet-based approach was observed. The fold-difference between the highest number of reads and lowest number of reads per sample was lower when using the ddPCR compared to conventional PCR (**Supplementary Table 9**). Further, the proportion of reads lost after trimming of low-quality reads was lower when using ddPCR (at 5 parasites/ μ L: 4.1-5.1% for ddPCR and 10.2-11.0% for conventional PCR). Coverage of markers was identical between the two methods.”

With our assay we obtained sequencing data in the majority of samples from Zanzibar at a density of >5 parasites/ μ L). Thus, sensitivity is on par with other published protocols (e.g., Tessema et al., 2020; LaVerriere et al., 2022 (though care needs to be taken due to different protocols for determining parasite densities). We have not systematically compared sensitivity of ddPCR-AmpSeq vs. conventional amplification, as this would require sequencing of a large number of samples around the limit of

detection and also more thoroughly consider read depth per sample (as this ultimately determines the number of markers with reads).

We recently applied the same approach for panels for *P. vivax* and *P. malariae*. This required minimal additional optimization (manuscripts will be published at a later time). Almost all markers that yielded product in a single-plex reaction worked in a multiplexed ddPCR, but much fewer in a conventional multiplexed reaction. We thus conclude that most conventional assays can easily be transformed into a ddPCR assay without extensive testing and optimization.

10) The article describes various inferences achieved from the 28 microhaplotypes analyzed, especially based on relatedness (IBD) calculation applied to polyclonal samples. For example, subtle variation in mean and fraction high IBD levels indicate malaria spread within shehias and households, malaria spread across the dry season, an admixture zone on Unguja island, and slightly higher island-mainland infection relatedness in recently traveled vs. non-travelling patients. Some inferences, e.g., patterns of connectivity between mainland and island populations appear more speculative in my appreciation. The discussion does well to summarize main inferences (e.g., ongoing parasite dispersal from the mainland to the islands as an obstacle to malaria elimination), although it was less clear how MOI and heterozygosity/diversity metrics fit the narrative and which/how patterns exactly are 'actionable' in this study.

RESPONSE: We agree that MOI and diversity are less informative in Zanzibar. We have elaborated on this in the discussion as follows (Page 33, lines 693-701):

~~“We found overall high genetic diversity on Zanzibar, indicative of ongoing transmission and frequent importation of infections. Although commonly believed that *P. falciparum* genetic diversity declines with decreasing transmission intensity¹⁸, we found overall high genetic diversity in Zanzibar with only subtle differences between districts, indicative of ongoing transmission and frequent importation. Similarly, a recent study in Eswatini found high parasite diversity despite being considered a low transmission area⁶⁸ and diversity might only decline once transmission is at very low levels⁶⁹. The effect of importation needs to be considered, particularly as countries approach elimination, and therefore MOI and genetic diversity may not be the most appropriate metrics to use as surrogate markers of transmission intensity in Zanzibar and other areas facing similar challenges.”~~

Within-host diversity indices of malaria infections might provide a useful metric for estimating transmission intensity in malaria-endemic regions, but its interpretation can be complicated in dynamic transmission settings with high rates of importation. Declining transmission has been associated with reduced parasite genetic diversity, less gene flow, and more substructure due to population bottlenecks and fragmentation of parasite populations. While overall a trend towards lower diversity in areas of low and focal transmission can be observed, differences in diversity are often minimal despite substantial differences in transmission intensity (Koepfli and Mueller, 2017). A recent study in Eswatini found a high level of parasite diversity despite being

considered a low-transmission area (Roh et al., 2019) and diversity might only decline once transmission is at very low levels (Nkhoma et al., 2013). This is also consistent with our study, as Zanzibar is considered a low transmission setting, however, diversity indices (MOI and also heterozygosity) are high. We conclude that this is in large part due to ongoing genetic exchange between the mainland and Zanzibar, which is in itself an interesting and important finding. At most, MOI, and heterozygosity point towards subtle differences in transmission, especially in northern Pemba, where both MOI and heterozygosity are lower, indicating more local transmission and less genetic exchange. This is further supported by anecdotal evidence that movement of people between study districts in Unguja and mainland is higher than between study districts in Pemba and the mainland.

My remaining comments are mainly cosmetic:

11) Line 45: *The number 518 is a bit misleading here given that it was 290 whose sequencing results sufficed for pop gen analysis.*

RESPONSE: We have changed this number accordingly. The abstract now reads (Page 3, lines 45-48):

“We **successfully** sequenced **290** ~~518~~ *P. falciparum* samples from 5 districts covering both main islands **of Zanzibar** using a novel, highly multiplexed droplet digital PCR (ddPCR)-based amplicon deep sequencing method targeting 35 microhaplotypes and drug-resistance loci.”

12) Line 46: *Perhaps ‘islands of Zanzibar’ instead of just ‘islands’.*

RESPONSE: We have changed this accordingly. The abstract now reads (Page 3, lines 45-48):

“We **successfully** sequenced **290** ~~518~~ *P. falciparum* samples from 5 districts covering both main islands **of Zanzibar** using a novel, highly multiplexed droplet digital PCR (ddPCR)-based amplicon deep sequencing method targeting 35 microhaplotypes and drug-resistance loci.”

13) Line 54: *Is ‘related’ here different than ‘highly related’ in line 51?*

RESPONSE: Yes, ‘related’ means all significantly related pairs as determined by hypothesis testing without any relatedness threshold, whereas ‘highly related’ refers to infection pairs at a relatedness level of IBD ≥ 0.9 . We have clarified this as follows (Page 3, lines 54-55):

“We identify a substantial fraction (2.9%) of **significantly** related **infection** parasite pairs between Zanzibar, and mainland Tanzania and Kenya, consistent with recent importation.”

14) Line 72: *Within this intro of malaria in Zanzibar, is it possible to add case numbers/year on mainland and islands?*

RESPONSE: We have integrated this information into the manuscript, however, we have added it to the Methods section, as we think it fits better there (Page 8, lines 170-173):

“In the two study years, ZAMEP reported a mean annual parasite incidence (API) per 1,000 population of 2.6 (approx. 4,000 cases per year; ZAMEP, unpublished data), as compared to an API of 106 reported by the mainland NMCP⁴⁶.”

15) Line 75: *I think as worded here the text is asserting that in lower-transmission intensity settings most malaria infections are asymptomatic, and most asymptomatic infections have parasite densities below the limit of detection of RDTs. Is this true as worded?*

RESPONSE: We have re-worded this section. What we wanted to say is that many cases are asymptomatic *and* that a substantial number of infections have densities below the limit of detection of RDTs. We have clarified this as follows (Page 4, lines 72-85):

“Since 2012, the Zanzibar Malaria Elimination Programme (ZAMEP) has implemented reactive case detection (RACD) to better target residual foci of transmission⁴. RACD includes screening by rapid diagnostic test (RDT) and treatment of household members of passively detected index cases at health facilities. **However, despite considerable efforts, malaria elimination in Zanzibar remains elusive. There are likely several reasons for this: firstly, despite the increasingly focal nature of residual transmission in Zanzibar^{3,5-7}, there is a substantial reservoir of asymptomatic infections in the community³. Many of these infections are low-density below the detection limit of RDTs^{2,8,9}, making it challenging to identify and treat them. Secondly, despite the implementation of strong vector control measures, there is ongoing local transmission due to residual vector capacity³. Lastly, Zanzibar is highly connected to mainland Tanzania where malaria transmission remains substantially higher in certain areas¹⁰⁻¹². Thus, even if local transmission has been reduced to very low levels, parasite importation through human travel might be a concern and obstacle to local elimination as long as the environment remains receptive¹³.**

16) Line 99: *‘Unmatched’ resolution wouldn’t be true unless you are referring only to non-WGS methods.*

RESPONSE: We agree and have changed accordingly. The Introduction now reads (Page 5, lines 102-104):

“Novel amplicon sequencing-based genotyping methods allow typing of parasites **at high** ~~with unmatched~~ resolution and sensitivity and provide a highly detailed picture of the parasite population structure^{27,28}.”

17) Line 102: *Can't SNPs be multi-allelic? Are you refereeing to number of polymorphic nucleotide positions within the analyzed marker, or number of alleles within the analyzed marker? (the former I'm not sure actually fits the meaning of 'multiallelic')*

RESPONSE: Yes, in rare cases, SNPs can be multi-allelic. Here, we referred to the *number of alleles within the analyzed marker* that exhibit multi-allelic rather than bi-allelic diversity. We agree that the wording was confusing and have changed this accordingly (Page 5, lines 107-109):

“By targeting short, highly diverse **microhaplotype loci that contain multiple SNPs and exhibit** ~~multiallelic loci (microhaplotypes)~~ rather than biallelic **diversity** ~~loci (e.g. SNPs)~~, discriminatory power and relatedness inference can be increased³⁰.”

18) Line 151: *It would be interesting to add these seaports as points on at least one plot. Or mention their names so readers can follow up. Also would be interesting to know if travel to Pemba is generally via Unguja or also commonly directly from mainland.*

RESPONSE: Figure 2A now includes the main seaports and the two airports. We have also added the following information to the methods section (Page 8, lines 157-164):

“The islands are highly connected to mainland Tanzania by two airports (international airport in Unguja) and ~~five~~ **three** main seaports (**Figure 2A**), ~~with additional frequent small boat traffic between the mainland and Zanzibar.~~ **In addition to this, there is frequent small boat traffic between the mainland and Zanzibar, often by traditional dhows. Travel to Pemba from mainland Tanzania can be through Unguja, mainly through the Malindi seaport, or directly from Tanga, a northern coastal region in mainland Tanzania, via either Mkoani or Wete seaports. Travelers can also reach Pemba from mainland Tanzania through the airport located at Chake Chake district.**”

19) Line 223: *What is >0.5 mismatch rate? Would >0.5 & <1 refer to a heterozygous call of a novel SNP?*

RESPONSE: Yes, a mismatch rate of >0.5 to the reference sequence indicates a potential novel SNP found on the dominant clone or a clone with a within-host frequency of >0.5 in an infection with MOI >1. If the sample is a single clone infection, we would expect a mismatch rate of 1. Cut-off criteria are introduced and discussed in

detail in an earlier publication (Lerch et al., 2017). We clarified this as follows in the Methods section (Page 11, lines 241-244):

“A mismatch rate of >0.5 to the reference sequence indicates a potential novel SNP found on the dominant clone or a clone with a within-host frequency of >0.5 at MOI >1 . For single clone infections a mismatch rate of 1 is expected.”

20) Line 243: Why did you not create the 1-2% strain ratios for 3D7/FCB?

RESPONSE: There is no additional knowledge gained by using the 1-2% strain ratios for 3D7:FCB for testing the limitation of the method, e.g., detectability and false-positive haplotype calls. The 3D7:HB3 ratios were readily available and are identical to the ones used in earlier publications where extensive investigations for cut-off criteria were undertaken (Lerch et al., 2017; Gruenberg et al., 2019). Including the same 3D7:HB3 ratios allowed us to compare performance of our new and the established assay.

21) Line 248: Why at least 2 loci?

RESPONSE: This is a common practice to protect against false-positive allele calls and therefore overestimation of MOI. Mean MOI does not change dramatically when using the loci with the single highest number of alleles (1.9 vs 2.1).

22) Line 338: Does ‘detection of alleles’ mean the same thing as % loci amplified (line 335)?

RESPONSE: Yes, it does. We have reworded this for consistency (Page 16, lines 383-385):

“Detection of alleles ranged from 81% to 94% of loci were successfully amplified in samples with as few as down to 10 parasites/ μ L (Supplementary **Figure 3D**).”

23) Line 339: Does ‘down to’ mean ‘at’?

RESPONSE: Yes, the reviewer is correct. We have changed the sentence as follows (Page 16, lines 383-385):

“Detection of alleles ranged from 81% to 94% of loci were successfully amplified in samples with as few as down to 10 parasites/ μ L (Supplementary **Figure 3D**).”

24) Line 346: I suggest to add the numbers for ‘still high’, else it sounds evasive I think.

RESPONSE: We have added the respective numbers. The Results now reads (Page 16, lines 392-393):

“At 10 parasites/ μ L, minority clone detection was still high at 40% (63.0% - 92.6%) and 20% frequency (48.1% - 66.7%).”

25) Line 349: Did lowering the detection limit to 0.1% lower precision of reference strain genotypes?

RESPONSE: Yes, with a detection limit of 0.1% false positive haplotypes calls are detected in control samples. This was also seen in earlier publications where thorough evaluation of a cut-off was done (Lerch et al., 2017; Gruenberg et al., 2019). Evaluation of four different amplicon analysis pipelines showed that all tools have reduced sensitivity and precision on samples with very low parasitemia or low read count (Early et al., 2019). Hence, lowering the detection limit to 0.1% is not recommended, even when detection of clones in control samples at this frequency is theoretically possible.

26) Line 355: Is there a histogram etc. in supplement to show parasite densities of the DBS sample set?

RESPONSE: Figure 1A shows the parasite densities of all DBS. However, we have added a histogram to the supplement (Supplementary Figure 4) that shows the parasite densities of all DBS for better illustration and highlighted the samples with data in ≥ 10 loci.

27) Line 385: What is inferred from the strong correlation between observed het. and global exp. het.? Could the authors add a comment on this here or in the discussion?

RESPONSE: The main message of this plot is to show that markers used are highly polymorphic in both, Zanzibar as well as globally, indicating that our panel can be used in sites outside Zanzibar/east Africa. We have made minor changes to the text on and added additional plots and H_E estimates to the supplement (Page 18, line 443-448):

“There was a moderate strong correlation between the expected heterozygosity based on a global dataset of *P. falciparum* genomes and the observed heterozygosity in Zanzibar, ($R^2 = 0.37$ ~~0.75~~; $P < 0.001$) (Figure 2C). Correlation was higher when sub-setting to samples from East Africa or Tanzania only, respectively (Supplementary Figure 6A & B). Overall, this shows that the microhaplotypes used are highly polymorphic both in Zanzibar and globally.”

Further, based on comment #51 by Reviewer 3, we have re-calculated H_E from a set of East African, Tanzanian and global samples using the newly released Pf7 dataset

from MalariaGEN that now contains over 20,000 samples. The correlation between the expected heterozygosity based on the global dataset of *P. falciparum* genomes and the observed heterozygosity in Zanzibar dropped when using the new Pf7 dataset ($R^2 = 0.37$, $P < 0.001$) compared to an older MalariaGEN release (MalariaGEN Plasmodium falciparum Community Project (2016) Genomic epidemiology of artemisinin resistant malaria *eLife* 5:e08714) used previously ($R^2 = 0.75$, $P < 0.001$). For some markers, the number of SNPs and therefore also the number of microhaplotypes has changed quite a bit, resulting in different H_E estimates for some. Likely MalariaGEN removed some low-quality SNPs and additionally the dataset is much larger.

28) Fig. 1B: Could you add DBS sample sizes for each bin or otherwise state this clearly somewhere in beginning of DBS results?

RESPONSE: We have added the DBS samples sizes for each bin in Figure 1B.

29) Fig. 2A: Possibly add mainland to map for context?

RESPONSE: We have updated Figure 2A, which now includes mainland Tanzania.

30) Fig. 2B: From the methods I understood that MOI was classified as an integer? Why are the sample points in the boxplot predominantly decimals while the medians appear to be integers 1 or 2? Sorry if I've missed something here.

RESPONSE: We thank the reviewer for spotting this error, MOI was indeed classified as an integer. When generating Figure 2B, we used the “geom_jitter()” function to highlight all the individual data points but this also adds a small amount of random variation to the location of each point. We have adjusted this parameter so that the jitter height is close to 0 and does not wrongly add decimals to the data points.

31) Fig 2B/2D: Why do sample size appear different between these 2 graphs?

RESPONSE: In Figure 2B, we highlighted the MOI of each individual infection as a point ($n=290$), whereas in Figure 2D the points reflect the heterozygosity of each microhaplotype ($n=28$), hence the difference. We have made this clear in the Figure legend:

“**B**, Multiplicity of infection (MOI) of 290 DBS samples across 5 districts. [...] **D**, Distribution of expected heterozygosity of the 28 microhaplotypes across 5 districts in Zanzibar.”

32) Line 422: Perhaps 'local transmission' could be reworded to something like 'locally restricted transmission'.

RESPONSE: We have reworded the sentence accordingly (Page 20, lines 486-487):

“Overall, this suggests frequent recombination of different parasite clones on both islands, but also locally **restricted** transmission, in particular, on Pemba Island.”

33) Fig 3. Very nice plot. You could possibly label Pemba/Unguja since these are referred to in text. Possibly could add a 2-plot version split by travel/no travel to supplement?

It's hard to see here if there is any subclustering among non-travellers around the 0,0 origin.

RESPONSE: Thank you. We have added labeling for Unguja and Pemba to the main Figure 3. We have also added two additional versions of the plot to the supplement highlighting either travelers (Supplementary Figure 10A) or non-travelers (Supplementary Figure 10B) to better visualize any sub-clustering. We have added the following to the Results (Page 20, lines 476-480):

“DAPC using the first 23 components explaining 49.8% variation of the original PCA (**Figure S6A&B**) inferred 3 distinct subpopulations by their district-level origin (**Figure 3, Supplementary Figure 10 A&B**), highlighting spatial clustering between districts, particularly separating samples from Micheweni and Chake Chake into more isolated populations.”

34) Line 456: Perhaps clarify “among non-travelers vs. travelers” = “among non-travelers vs. among travelers”.

RESPONSE: We have made this change to make this clearer. The Results now reads (Page 22, lines 522-524):

“We also found a 1.3-fold higher mean IBD among non-travelers vs. **among** travelers to mainland Tanzania and Kenya (0.04 vs. 0.03, permutation test, $P = 0.003$), possibly due to subtle population structure differences between Zanzibar and the mainland.”

35) Line 531: Hard to see how these correspond well. To me it looks like the western region of Unguja near port of entry has higher IBD to mainland, as well as southern Pemba, though not the next adjacent district in Pemba which has low IBD to mainland but similar travel affinity based on the questionnaires.

RESPONSE: In this sentence we specifically referred to Unguja, where travel and mean IBD/proportion of related infections to the mainland seems to correspond fairly well. It also corresponds well for the northern districts of Pemba (lower relatedness and less travel reported). The reviewer is correct that for the southern district in Pemba (Mkoani) where relatedness is higher, but reported travel to the mainland is low, this seems not to be the case. However, Mkoani is the main port of entry when travelling from/through Unguja, hence probably has the highest “connectivity” to Unguja and thus also indirectly to the mainland. We have rephrased this accordingly (Page 27, lines 598-606):

“Stratifying by the 5 districts, we find clear differences in their mean IBD and the proportion of related infections to the mainland (Supplementary **Figure 16A&B**), corresponding well for districts in Unguja with more frequently reported travel to the mainland, and northern districts of Pemba with less frequently reported travel to the mainland (Supplementary **Figure 16C**) and higher reported travel of any household member of infected individuals from Unguja (**Figure 7A&B**, Supplementary **Figure 16D**). The southern district of Pemba, Mkoani, shows higher relatedness but less frequently reported travel to the mainland. However, Mkoani is the main port of entry when travelling from Unguja or Tanga on mainland Tanzania, possibly explaining this phenomenon.”

36) Fig. 7: What are the sample size for each geographical node in this analysis? How much does # of pairwise comparisons differ among the edges and may this distort inference? E.g., do you get the same results when balancing sampling randomly or applying some sort of bootstrapping approach?

RESPONSE: This is a valid concern. As the table below shows, the number of samples from each location varies and so does the number of pairwise comparisons between two locations and within each location. Consequently, the variances of location-level (district-level) estimates, such as a mean relatedness or a proportion of significantly related pairs, are different and so are their levels of uncertainty. We have addressed this issue by performing a permutation test, which preserves the dependence structure of the data (pairs of samples are not independent) while providing statistical inference that takes into account the number of pairwise comparisons. We used a resampling method, generating null distributions (H_0 : no geographical structure) for district-level estimates by reshuffling location labels of the samples, followed by calculating district-level p-values for this hypothesis.

	n samples	n pairs	Mean IBD	P value	Proportion related	P value
Micheweni	57					
Kilifi	35	1,995	0.020	1	0.022	0.999
Muheza	10	570	0.031	0.474	0.026	0.924
Mkuzi-Muheza	35	1,995	0.018	1	0.017	1
Morogoro	15	855	0.022	0.996	0.022	0.992

Nachingwea	20	1,140	0.027	0.864	0.029	0.931
Chake Chake	43					
Kilifi	35	1,505	0.015	1	0.017	1
Muheza	10	430	0.018	0.999	0.019	0.988
Mkuzi-Muheza	35	1,505	0.020	1	0.025	0.996
Morogoro	15	645	0.018	0.999	0.020	0.993
Nachingwea	20	860	0.021	0.999	0.019	0.999
Mkoani	20					
Kilifi	35	700	0.024	0.963	0.027	0.930
Muheza	10	200	0.021	0.959	0.010	0.994
Mkuzi-Muheza	35	700	0.027	0.846	0.037	0.518
Morogoro	15	300	0.022	0.951	0.037	0.554
Nachingwea	20	400	0.030	0.548	0.045	0.238
Magharibi	121					
Kilifi	35	4,235	0.024	0.999	0.026	0.999
Muheza	10	1,210	0.024	0.979	0.035	0.666
Mkuzi-Muheza	35	4,235	0.026	0.992	0.040	0.165
Morogoro	15	1,815	0.024	0.994	0.039	0.397
Nachingwea	20	2,420	0.028	0.854	0.033	0.861
Kusini	49					
Kilifi	35	1,715	0.024	0.999	0.028	0.979
Muheza	10	490	0.022	0.979	0.033	0.719
Mkuzi-Muheza	35	1,715	0.027	0.953	0.039	0.354
Morogoro	15	735	0.022	0.994	0.038	0.469
Nachingwea	20	980	0.022	0.999	0.024	0.985

37) Was WGS data from localities on the mainland closer to the islands (e.g., coastal Pwani) not applied in IBD analyses because they were not available from WGS or because travel there is not common based on the questionnaires? Or other reasons that area is less relevant...?

RESPONSE: Unfortunately, there were no WGS data available from localities closer to the islands (e.g., Pwani). However, the localities included here are relevant for our study as people reported recent travel to all of them.

38) Fig. 9B: Again couldn't hurt to add island labels.

RESPONSE: We have added island labels to Figure 9B and have slightly changed the text of the legend of Figure 9. It now reads (Page 32, lines 684-685):

“Map showing drug resistant isolates on Unguja (left) and Pemba (right) a map of Zanzibar.”

39) Line 625: *Aside from spatially correlated dispersal, do any metrics of this study allow inference of transmission intensity differences? E.g., between Pemba and Unguja?*

RESPONSE: Supplementary Figure 7 highlights differences in MOI and heterozygosity by islands, travel history and year. We find significant differences in MOI (Pemba 1.69 vs. Unguja 1.95, $P = 0.021$) and heterozygosity (Pemba 0.70 vs. Unguja 0.73, $P = 0.012$) between the islands. Additionally, we also see significant differences in MOI between non-travelers and travelers (non-traveler 1.72 vs. traveler 2.13, $P = 0.008$). We have added the following to the results (Page 19, lines 451-454):

“We also investigated heterozygosity and MOI by travel history, year, and island and found significantly higher population diversity and MOI on Unguja than Pemba (MOI 1.95 vs. 1.69, $P = 0.021$; H_E 0.73 vs. 0.70, $P = 0.012$) island, as well as higher MOI in individuals who reported recent travel (2.13 vs. 1.72, $P = 0.008$) (Supplementary Figure 7A-F).”

Within-host diversity indices of malaria infections might provide a useful metric for estimating transmission intensity in malaria-endemic regions, but its interpretation can be complicated in dynamic transmission settings with high rates of importation. Declining transmission has been associated with reduced parasite genetic diversity, less gene flow, and more substructure due to population bottlenecks and fragmentation of parasite populations. While overall a trend towards lower diversity in areas of low and focal transmission can be observed, differences in diversity are often minimal despite substantial differences in transmission intensity (Koepfli and Mueller, 2017). A recent study in Eswatini found a high level of parasite diversity despite being considered a low-transmission area (Roh et al., 2019) and diversity might only decline once transmission is at very low levels (Nkhoma et al., 2013). This is also consistent with our study, as Zanzibar is considered a low transmission setting, however, diversity indices (MOI and also heterozygosity) are high. We conclude that this is in large part due to ongoing genetic exchange between the mainland and Zanzibar, which is in itself an interesting and important finding. At most, MOI, and heterozygosity point towards subtle differences in transmission, especially in northern Pemba, where both MOI and heterozygosity are lower, indicating more local transmission and less genetic exchange. This is further supported by anecdotal evidence that movement of people between study districts in Unguja and mainland is higher than between study districts in Pemba and the mainland.

Reviewer #3:

Major comments

40) For me the most interesting aspect of this paper is that there appears to be evidence of local transmission in north Pemba, but this is less evident in Mkoani or on Unguja (as evidenced by Figure 3 and Figure S8A/B). An alternative explanation for these results, however, could be that the sampling framework was different in north Pemba, for example many more samples from the same households (RACD samples) were taken in Micheweni and Chake Chake than in other districts. Indeed, on lines 694-695 it is stated that "sampling of parasites was linked to routine RACD by ZAMEP and hence not spatially uniform across geographic locations", but details are not given. The authors should provide a breakdown of samples from the same vs different households by district and year so this alternative explanation could be ruled out.

RESPONSE: We have provided a breakdown of the 290 samples successfully genotyped. The proportion of samples from within the same households is not higher in north Pemba (Micheweni and Chake Chake; highlighted in orange color) districts than in others (it is lower). Hence, this alternative explanation can be ruled out. We have added this information as follows:

(Page 17, lines 411-414)

"These samples are representative of all 5 districts from which the original sample set was collected, with no clear differences in the proportion of samples from the same vs different households between Unguja and Pemba (Supplementary Table 11 and 12)."

(Page 36, lines 783-787)

"Sampling of parasites was linked to routine RACD by ZAMEP and hence not spatially uniform across geographic locations. Among the 290 samples successfully genotyped, the proportion of samples from within the same households varies from 25% to 66%, with no clear differences between Unguja and Pemba."

Regarding the study design: The procedures were standardized across the entire study area, and the survey was embedded into the procedural routine of the RACD system in Zanzibar. The lower number of samples collected in Mkoani could be attributed to the district reporting fewer cases due to the area's low malaria prevalence. To account for this, we included two districts, Mkoani and Chake Chake, both of which have low malaria prevalence in the southern region of Pemba, instead of selecting just one of the two districts. This is also the reason why we have three districts in Pemba and two in Unguja.

District	n	Same households (%)	Different households (%)
Pemba			
Micheweni 2017	45	24 (53.3%)	21 (46.7%)
Micheweni 2018	12	4 (33.3%)	8 (66.7%)
Chake Chake 2017	27	9 (33.3%)	18 (66.7%)
Chake Chake 2018	16	4 (25.0%)	12 (75.0%)
Mkoani 2017	11	4 (36.4%)	7 (63.6%)
Mkoani 2018	9	5 (55.6%)	4 (44.4%)

Unguja

Magharibi 2017	44	24 (54.5%)	20 (45.5%)
Magharibi 2018	77	51 (66.2%)	26 (33.8%)
Kusini 2017	27	11 (40.7%)	16 (59.3%)
Kusini 2018	22	12 (54.5%)	10 (45.5%)

41) *I am somewhat sceptical that reliable measures of IBD can really be determined with just 28 genetic markers. For example, the inset of Figure 4A shows a perhaps surprising number of sample pairs with mean IBD between 0.75 and 0.9. This would suggest these are not clonal pairs but also that they are more highly related than, for example, sibling pairs. This could be evidence of multi-generational inbreeding but might also be an artefact of attempting to estimate genome-wide IBD with a small number of markers. Figure 4 of the Dcifer paper (reference 42) suggests confidence intervals for relatedness measures when using < 50 loci are likely to be very high. Moreover, even in a study of a highly-related South American populations using whole genome data (<https://pubmed.ncbi.nlm.nih.gov/36542676/>), very few sample pairs have IBD between 0.75 and 0.9. The authors should comment on the somewhat surprising nature of the IBD estimates in their study, for example by stating whether they think they have evidence of multi-generational inbreeding or whether they think this is an artefact*

RESPONSE: This is a valid point. We have divided our response into two parts:

1) We have assessed the statistical power of our 28 microhaplotype panel using two simulation schemes: (a) no genotyping errors; true MOI and population allele frequencies used as inputs to dcifer and (b) including genotyping error with fixed error model parameters; MOIs and allele frequencies estimated from these data (see Figure below). The false positive rates (FPR) were below the nominal significance level $\alpha=0.05$. As expected, power was greater for higher values of relatedness, and higher MOI led to lower power. For our panel of 28 microhaplotypes, the power to detect siblings ($r = 0.5$) in a pair of infections with MOI of 2 was 0.91, and 0.72 in a pair of infections with MOI of 3. We acknowledge that the power drops at higher MOIs, however, this is barely an issue with our data from Zanzibar as MOI between most pairs is rarely above 2-3 (2/1,703 significantly pairs are MOI > 3 for both). We also acknowledge that the power is somewhat lower than for instance for the 91 microhaplotype panel presented in the Dcifer paper, however, power is likely still higher than for instance the biallelic 101 SNP barcode panel (also used in the Dcifer paper).

We have added the simulations to the Methods section (Page 13-14, lines 315-324):
 “We have calculated the statistical power and false-positive rate (FPR) of our 28 microhaplotype panel at different relatedness levels using two simulation schemes described elsewhere⁴²: (a) no genotyping errors; true MOI and population allele frequencies used as inputs to *dcifer*, and (b) including genotyping error with fixed error model parameters; MOIs and allele frequencies estimated from these data. The FPR was below the nominal significance level $\alpha=0.05$ (**Supplementary Figure 2**). As expected, power was greater for higher values of relatedness and higher MOI led to lower power. For our panel of 28 microhaplotypes, the power to detect siblings ($r=0.5$) in a pair of infections with MOI of 2 was 0.91, and 0.72 in a pair of infections with MOI of 3; in our study MOI between most infection pairs was rarely above 2-3.”

2) As a first step in our assessment of relatedness, we used a [default] setting where only a single pair of strains between 2 infections ($M=1$) regardless of MOI can be related. However, *Dcifer* also allows for multiple pairs of strains to be related between two infections ($M \geq 2$). When we allow that multiple pairs of strains may be related, *Dcifer* produces a corresponding number of estimates—one for each pair. But a large number of diverse loci is needed; otherwise, there is a lot of variation in the individual estimates. However, their sum r_{total} can be estimated more accurately even with a lower number of loci. Hence, we can have three different scenarios: a) both samples have MOI = 1, b) both samples have MOI > 1, and c) one sample has MOI = 1 and the second sample has MOI > 1.

a) MOI = 1 for both samples. Apart from multi-generational inbreeding, genotyping errors might play a role here. As can be seen in Supplementary figures S.2, S.11, and S.12 of *Dcifer* paper where results of simulations with “genotyping errors” are

presented, for truly clonal samples, these errors might break the “perfect correspondence” consequently resulting in relatedness estimates that are somewhat lower than 1. The wide confidence interval concern is certainly also valid; to check that, we looked at the 95% CI for such sample pairs supplied by Dcifer likelihood-ratio-based inference, which accounts for the number of markers in the panel. Interestingly, most such intervals are contained within the (0.5, 1) interval, and only a small fraction extends below 0.5. For the 290 Zanzibar samples, we found only one pair with a wide 95% CI, out of 18 pairs with COI of 1-1 and an estimate between 0.75 and 1. The estimate for that pair is 0.754, with 95% CI = [0.216, 0.984].

b) $MOI > 1$ for both samples. For these pairs of polyclonal infections, we also consider the possibility that multiple pairs of individual parasite strains between two infections are related (the initial analysis procedure constrains relatedness between two infections to a single related pair of strains). To explore this possibility, we used dcifer to estimate the number of related pairs of strains between these infections and found that in many such cases there is evidence of multiple related strain pairs and high overall relatedness, which would indicate either coinfection or multiple inoculations with the same clones. With multiple related strain pairs, their levels of relatedness can be different, e.g. one pair being clonal and another at or below 0.5.

For all significantly related pairs of samples with $MOI \geq 2$, we estimated M' (the number of related strain pairs) and r_{total} (overall relatedness). For these samples, r_{total} ranged between 0.1 and 2 and there were 173 pairs of samples for which r_{total} exceeded 1. There are 175 sample pairs with IBD between 0.75 and 0.9. We found that in many such cases there is evidence of multiple related strain pairs. For these sample pairs, r_{total} ranged between 0.754 and 1.72. There were 90 pairs with a r_{total} greater than 1; with an estimated MOI of ≥ 2 and $M \geq 2$ for all such pairs. The proportion of polyclonal samples for these highly related pairs is about the same as for all samples, and MOI is 2 and 3 (mostly 2) so it is not a factor driving the overall relatedness up - it's the actual high connectivity.

We also looked into these highly related pairs ($IBD \geq 0.75$) where multiple strains are related ($M \geq 2$) and their geographical locations/distributions (i.e., household, RACD cluster, Shehia; see below). Among highly related pairs ($IBD \geq 0.75$) with $M \geq 2$ the proportion to find those within the same group was much higher than for the total number of pairs found within, indicating local transmission.

	Total number of pairs	Number of highly related multiple strain pairs ($IBD \geq 0.75$ and $M \geq 2$)
Within same Household	114/41905 (0.27%)	15/134 (11.19%)
Within same RACD cluster	233/41905 (0.56%)	24/134 (17.91%)
Within same Shehia	1474/41905 (3.52%)	76/134 (56.72%)

c) MOI = 1 for one sample and MOI > 1 for another sample in the pair. In regard to Dcifer relatedness estimation, this case is similar to (a), with added complexity being reflected in a little more uncertainty.

Thus, highly related does not necessarily imply inbreeding when using Dcifer (with amplicon data) and relatedness inferences when using Dcifer differ quite a bit from relatedness inferences using WGS data (or even MIP data), as we are not restricted to just the major clone.

42) Also on the topic of IBD, I am sceptical that the apparently highly related samples between Zanzibar and mainland Africa shown in Figures 6, 7 and 8 are truly highly-related and not apparently highly related by chance due to the low number of genetic markers used. One thing that would give the reader more confidence would be showing that there are no highly related sample pairs between samples from Zanzibar and those from elsewhere in the world, e.g. from Asia. For example, Figure 9B could also include pairwise IBD with samples from Asian countries.

RESPONSE: The response to comment #41 applies here as well. 95% CIs based on likelihood ratio tests (which actually might be conservative since they are two-sided) indicate that most of these samples are indeed highly related. Really wide 95% CIs (like the pair we mentioned in the previous response) arise from matching alleles being common alleles in the population, meaning the matching is more likely to have happened by chance - but that seems to be the case only in a small number of pairs: among all pairs with estimates above 0.75, 1 out of 53 for MOI = 1-1 and 3 out of 295 for all MOI.

We also have repeated the pairwise IBD analysis and included the samples from Asia. Mean pairwise IBD is indeed lowest between the samples from Zanzibar and Asia. We have included pairwise IBD with samples from Asian countries and added mean pairwise IBD values to **Figure 8B** to make this clearer. We have also updated **Supplementary Table 16** showing mean pairwise IBD and proportion of related infections (without any IBD threshold and with a threshold of IBD ≥ 0.25) accordingly.

Minor comments

43) The gene CRT was not included in panel, despite AS-AQ being used for treatment (AQ has been associated with K76T in some studies). I think many readers might be surprised to see the omission of this gene from the panel, so a short description of why this was not included might be helpful.

RESPONSE: The reviewer is correct. *pfCRT* was not included as we did not get satisfactory amplification. We added this to the Methods (Page 15, lines 362-363):

“Though associated with AQ resistance⁶², *pfcr* was not typed as no amplification product could be obtained with our assay.”

44) *The coverage profile of different amplicons (Figure 1C) shows large variation. As an example the average number of reads for c27 appears to be over 100x greater than that for t01. This might not be immediately obvious to many readers because the y-axis of this figure is on a log-scale. It is recommended that either the figure be recreated with a non-log scale, or that at least it is clearly highlighted that a log scale is used which might make the variation between amplicons appear to be lower than is actually the case.*

RESPONSE: We have highlighted that a log10-scale was used in the Figure legend. We have also added this to the y-axis title. It now reads (Page 18, lines 433-435):

“**C**, Average number of reads per target per sample. The median (bars) and interquartile range (error bars) are shown. Colored by microhaplotypes (light blue) and drug resistance loci (dark blue). **Note that the y-axis is on a log10-scale.**”

45) *The large variation in coverage between amplicons seen in Figure 1C should be highlighted in the main text, together with any suggestions as to how more even coverage might be achieved (e.g. through primer balancing). In particular, I don't think the statements "high level of uniformity in the average number of reads per target" on lines 366-367 and "relatively even amplification" on line 730 can be supported and should be removed.*

RESPONSE: We reworded this and added details to the discussion. The Results (Page 17, lines 420-423) and Discussion now read as follows (Page 37, lines 820-823):

(Page 17, lines 420-423)

“High coverage (median 98.6%) in samples with ≥ 10 parasites/ μL was achieved (**Figure 1B**). **However, with a high level of uniformity in the average number of reads per target in the 290 samples with ≥ 10 loci showed considerable variation among some of the markers (Figure 1C).**”

(Page 37, lines 820-823)

“The ddPCR-amplicon sequencing approach yielded high read numbers across 35 loci ~~with relatively even amplification~~ and **overall** coverage of all markers in samples as low as 10 parasites/ μL . **More even amplification might be achieved by varying primer concentrations (i.e., primer balancing) to account for amplification rate differences among loci.**”

46) In Table S1, the median and IQR of reads per locus are given, but I think the range might also be useful information here, or at least 5% and 95% quantiles.

RESPONSE: We have replaced IQR with range, for both, the median reads, and the median reads per locus. We also identified and corrected a minor error for one of the median values in the column “Median reads per locus (IQR)” in the last row. The value was 494 where it should have been 472.

47) In Figures S2B and S2D, coverage needs to be defined. Is the coverage at 1X, 10X or something else?

RESPONSE: By “Coverage (%)” we refer to the percentage of loci with reads for each sample. For instance, 100% coverage indicates that 35/35 loci had reads and were thus successfully amplified. Coverage was determined based on the number of targets with 10 or more reads. We have added a more detailed description to the figure legend of Supplementary Figure 3 in the supplement. The Figure legend now reads:

“Boxplot summarizing the coverage of microhaplotype loci and drug resistance targets by parasite density. Coverage was determined based on the number of targets with 10 or more reads (e.g., 100% indicates 35/35 loci with reads).”

Similarly, we added this detail to Figure 1 in the main text. The Figure legend now reads (Page 18, lines 430-432):

“Boxplot summarizing the coverage of microhaplotype loci and drug resistance targets by parasite density in 518 DBS samples from Zanzibar. Coverage was determined based on the number of targets with 10 or more reads (e.g., 100% indicates 35/35 loci with reads).”

48) In Figure S2B it appears that some points appear to be slightly greater than 100%. Is that true, and if so, how can coverage be greater than 100%?

RESPONSE: The reviewer is correct, that some points appear to be slightly greater than 100%, when they in fact should be at the 100% mark. This was caused when using the “geom_jitter()” function in R to generate the figure. This function adds a small amount of random variation to the location of each point to make data points of identical value better visible. We have adjusted this parameter so that the jitter height is close to 0 and does not wrongly plot the data points in Supplementary Figure 3B. Similarly, this was also an issue in Figure 1B in the main text, hence this was corrected accordingly as well.

49) *MOI is defined as "the highest number of alleles detected by at least two of the 28 microhaplotype loci". As such MOI for each sample should be an integer, but in Figure 2B the values are not integers. This requires an explanation.*

RESPONSE: We thank the reviewer for spotting this error, MOI was indeed classified as an integer. When generating Figure 2B, we used the “geom_jitter()” function to highlight all the individual data points but this also adds a small amount of random variation to the location of each point. We have adjusted this parameter so that the jitter height is close to 0 and does not wrongly add decimals to the data points.

50) *Some points on Figure 2B have MOI < 1. Is this a mistake? If not, what is the biological meaning of a sample with MOI less than 1?*

RESPONSE: See response above.

51) *For Figure 2C, the authors might want to consider showing estimates of heterozygosity from a set of east African or Tanzanian samples, as presumably this should have even higher correlation with observed heterozygosity than a global set of samples?*

RESPONSE: This is a fair point. As presumed by the reviewer, correlation was much higher using estimates of heterozygosity from a set of East African ($R^2 = 0.57$, $P < 0.001$) or Tanzanian samples ($R^2 = 0.59$, $P < 0.001$) compared to the global dataset. We have re-calculated H_E from a set of East African, Tanzanian and global samples using the newly released Pf7 dataset from MalariaGEN that now contains over 20,000 samples. The correlation between the expected heterozygosity based on the global dataset of *P. falciparum* genomes and the observed heterozygosity in Zanzibar dropped when using the new Pf7 dataset ($R^2 = 0.37$, $P < 0.001$) compared to an older MalariaGEN release (MalariaGEN Plasmodium falciparum Community Project (2016) Genomic epidemiology of artemisinin resistant malaria *eLife* 5:e08714) used previously ($R^2 = 0.75$, $P < 0.001$). For some markers, the number of SNPs and therefore also the number of microhaplotypes has changed quite a bit, resulting in different H_E estimates for some. Likely MalariaGEN removed some low-quality SNPs and additionally the dataset is much larger.

The main message of this plot, however, is to show that markers used are highly polymorphic in both, Zanzibar as well as globally. We have made minor changes to the text and added additional plots and H_E estimates to the supplement (Page 18, line 443-448):

“There was a **moderate** ~~strong~~ correlation between the expected heterozygosity based on a global dataset of *P. falciparum* genomes and the observed heterozygosity in Zanzibar, ($R^2 = 0.37$ ~~0.75~~; $P < 0.001$) (Figure 2C). **Correlation was higher when sub-setting to samples from East Africa or Tanzania only, respectively**

(Supplementary Figure 6A & B). Overall, this shows that the microhaplotypes used are highly polymorphic both in Zanzibar and globally.”

52) I presume the statement "Samples in Micheweni share a high proportion between transmission seasons" on line 453 is based on the Micheweni 2017 vs Micheweni 2018 cells in Figure S8C/D. It might be helpful to the reader to refer to that figure here, and maybe put the mean IBD (0.101) and/or Proportion of related infection pairs (0.107) results in the main text.

RESPONSE: We thank the reviewer for spotting this. We indeed refer to Figure S8 C and D (now Supplementary **Figure 12**), however, we did not refer to this in the main text. We have added this information and also added the mean IBD and proportion of related infection pairs to the main text. The Results now read (Page 22, lines 515-522):

“When we stratified the districts by year, we found that mean IBD and proportion of related infections in Micheweni and Chake Chake districts were higher in samples from 2017 than 2018 (**Supplementary Figure 12C&D**). However, we observed distinct patterns for the two districts. Samples in Micheweni share a high proportion between transmission seasons (**mean IBD = 0.101, permutation test, $P < 0.0001$; proportion related = 10.7%, permutation test, $P < 0.0001$**), indicating that a proportion of the parasite population is maintained through the dry season and contributes to malaria cases the following season.”

53) The statement "This was still true when we excluded all sample pairs where both individuals reported recent travel" on lines 484-485 states something that presumably would be expected; samples from patients that have travelled are presumably less likely to be related and therefore removing these would mean the samples that remain are expected to more highly related. An arguably more interesting analysis here would be to look at the levels of relatedness only among pairs of samples where one (or both) have travelled. If these showed significantly lower relatedness, that would presumably be of interest? Perhaps such an analysis could be added to Table 1?

RESPONSE: We have investigated the levels of relatedness (mean IBD and proportion of highly related sample pairs with $IBD \geq 0.9$) among pairs of samples at different levels (i.e., shehia, RACD cluster, and household) with (1) no travel, (2) where one travelled, and (3) where both travelled. Mean IBD is always higher within those groups than between groups. At all levels, mean IBD is highest within groups for pairs reporting no travel and lowest where one of the two reported travel. This is expected, as no travel and also co-travel are likely scenarios of being infected at the same time/place, whereas in pairs where only one traveled, there is a chance that the individuals got infected at different times/places. Pairs where one or both travelled show lower proportion of highly related sample pairs and mean IBD than when looking at all pairs. Similarly, pairs without travel show the highest proportion of highly related sample pairs and mean IBD.

We have adjusted Table 1 added the following to the Results (Page 23, line 551-553):
“This was also reflected by higher mean IBD within than between groups (Table 1). The levels of relatedness among pairs of samples were lowest where one travelled and highest among pairs without any travel. ~~This was still true when we excluded all sample pairs where both individuals reported recent travel.~~”

54) *On lines 190 and 256 it is stated that "All samples were sequenced in duplicate", and it is also stated that "A total of 290 (56%) DBS samples with data 356 in ≥ 10 loci for both replicates and a minimum of 10 reads per marker were included in further analyses", but I could see no details of how final results for each sample were arrived at. E.g. was there always perfect concordance between duplicate pairs and if not how were discrepancies resolved? This should be explained in the Methods section.*

RESPONSE: A haplotype of a sample was only called if it was present in both replicates. True haplotypes should be detected in both replicates, unless the sequence depth is not sufficient for detecting a minority clone in one of the replicates.

We have added the following to the methods (Page 11, line 247-248):

“To further reduce false-positive haplotype calls, each haplotype had to be confirmed by both replicates of a sample.”

55) *On line 318, it might be helpful to explain why mutations R539T and I543T have been singled out, rather than, for example, the more common C580Y mutation or mutations seen recently in Africa such as R561H, A675V or C469Y*

RESPONSE: Our amplicons did not cover all of the *pfk13* gene, and thus mutations R561H, A675V or C469Y were not covered. Our main focus when designing our panel were the highly diverse microhaplotypes, with markers of drug resistance being a secondary aim. Mutations R561H, A675V or C469Y had not been described in Africa when we designed our panel. We have since developed protocols for MinION sequencing of the full *pfk13* gene and sequenced isolates from Zanzibar, these results will be published in a separate manuscript we are currently preparing.

56) *On lines 320-321, it is unclear whether the phrase "samples genotyped" at the end of the sentence includes or excludes samples for which genotypes could not be determined (i.e. which have missing genotypes) which is an important distinction. This should be clarified.*

RESPONSE: We agree and have further clarified this. The Methods now reads (Page 15, lines 364-366):

"The mutation frequencies in the population were calculated as the number of samples that contained the mutation over the total number of samples **successfully** genotyped **at each locus**."

57) On line 548 "Three pairs from two individuals with IBD >0.5" are mentioned. Does this refer to the two red lines on Figure 7C? If so, it might be helpful to the reader to state this explicitly. Also, should this read "Two pairs from three individuals"?

RESPONSE: There are two individuals from Zanzibar that are IBD >0.5 to three infections from mainland Tanzania and Kenya; one individual from Zanzibar is related to two other infections and one individual is related to one infection. Hence, it is indeed two individuals from Zanzibar that have pairwise IBD >0.5 to three infections from the mainland. Thus, it should be three red lines that are highlighted. We have changed the settings for the color gradient to visualize this better and also provide a zoomed-in version of Zanzibar. IBD of one of the three pairs is just slightly higher than >0.5 (IBD = 0.516), thus the third line is lighter red than the other two. We have also slightly changed the wording to make this clearer and decided to move Figure 7C to the supplement (now Supplementary Figure 17), so that Figure 7A and B can be shown at larger scale in the main text. The Results now reads (Page 28, lines 622-625):

"Three pairs ~~from two individuals~~ with IBD >0.5 were identified (**Supplementary Figure 17**) **between three mainland samples and two individuals from Zanzibar (dark-red lines; both from Magharibi district Figure 7C)**; one of the two individuals reported recent travel to mainland Tanzania."

We have also added the following to highlight the differences in relatedness to the mainland between the two islands (Page 27, lines 606-609):

"**Mean IBD and the proportion of related infections was significantly higher between the mainland and Unguja than between the mainland and Pemba (mean IBD: 0.025 vs. 0.021, permutation test, $P = 0.0001$; proportion related: 0.034 vs. 0.023, permutation test, $P < 0.0001$; Supplementary Table 15).**"

58) On line 567-568 it is stated that "Zanzibar samples are more closely related to samples from east Africa and central Africa than to samples from west Africa". Which countries are considered east and central here? And is this statement based on formal analysis or inferred from Figure 8B? My reading of this figure is that IBD measures look somewhat higher from Kenya and Tanzania (which presumably are considered east Africa), but that IBD measures from countries that might be considered central such as Malawi or DRC don't look particularly higher than IBD measures from west Africa. Perhaps the phrase should be changed to "Zanzibar samples are more closely related to samples from east Africa than to samples from elsewhere in Africa" or similar?

RESPONSE: Malawi, Kenya, and Tanzania are considered east Africa; DRC is considered central Africa. The Gambia, Guinea, Mali, Ghana, and Nigeria are

considered west Africa. The statement is inferred from Figure 8B. We have added pairwise IBD for samples from Asia as well, and mean IBD is now indicated in Figure 8B (and in Supplementary Table 15). There clearly appear to be differences in relatedness between Zanzibar and east/central Africa, west Africa, and southeast Asia. Samples from Zanzibar show higher mean IBD to infections from east and central Africa (range: 0.026 – 0.029) than to infections from west Africa (range: 0.017 – 0.019). Mean pairwise IBD to infections from Asia is lowest (range: 0.004 – 0.008). We have changed the phrase to make this clearer. The Results now reads (Page 29, lines 642-646):

“Consistent with the DAPC results, we found that the Zanzibar samples appear to be more closely related to samples from east Africa (i.e., Kenya, Malawi, and Tanzania) and central Africa (i.e., DRC) than to samples from west Africa and southeast Asia (Figure 8B, Supplementary Table 15)., and they are most related. However, relatedness within Zanzibar was higher than between any other country (Figure 8B, Table S7).”

59) On lines 597-600 it would be helpful to quantify prevalence of triple mutant and N86-184F haplotypes (% and n).

RESPONSE: We have added this information. The Results section now reads (Page 31, lines 676-679):

“All parasites carried mutations in the *pfdhfr* gene, with the common *pfdhfr* triple mutant (51I-59R-108N) being particularly prevalent (66/74; 89.2%). We also observed a high number of the *pfmdr1* N86-184F haplotype (38/74; 51.4%).”

60) On lines 651-653 it is stated that "Targeting such areas with a combination of reactive focal MDA and reactive vector control is expected to have a complementary effect", but it is not clear whether such approaches are actively being considered in Zanzibar. If possible, I think it would be useful to expand on this, and also give an indication of whether the results presented here are actively being considered for future control and elimination efforts, e.g. by ZAMEP. I feel the manuscript might be of greater interest to many readers if it was clear that this analysis had driven a change in policy.

RESPONSE: This is a good point, and we are happy to add a few sentences on this. This study and previous work by us (Stuck et al., 2020; Grossenbacher et al., 2020; van der Horst et al., 2020) on the effectiveness of the RACD system in Zanzibar has led ZAMEP to consider alternative approaches (i.e., rfMDA and RAVC). They have now commissioned an SOP and plan for piloting these approaches and a process has been initiated to pilot test this approach. We have added this to the Discussion (Page 35, lines 742-745):

“The results of our study and previous findings from the RADZEC project^{2,4,8} have led ZAMEP to actively consider these approaches to enhance local elimination efforts.

ZAMEP has initiated the process of developing standard operating procedures and piloting for reactive focal MDA."

61) On lines 660-663 it is stated that "We identified a substantial number of infection pairs at the relatedness level of half-siblings or more ($IBD \geq 0.25$), likely representing imported cases and providing evidence for recent, and most likely ongoing, genetic exchange between Zanzibar and the mainland". If IBD measures ≥ 0.25 determined in this study are truly evidence for recent genetic exchange between Zanzibar and the mainland, Figure 8B would suggest there has been recent exchange from across Africa. The authors should comment on whether they think this truly is the case.

RESPONSE: Overall mean pairwise relatedness is lowest between the samples from Zanzibar and countries in Asia. Similarly, within Africa, West African countries show lower mean IBD to Zanzibar than East African countries (including DRC). The same pattern is true for the proportion of related infections. When looking at all pairs with $IBD \geq 0.25$, we again see a similar pattern, with the exception of The Gambia, showing similar proportions like East African countries (see Figure 8B and Supplementary Table 15). The proportion of related infections with $IBD \geq 0.25$ is very low (or zero) with countries from southeast Asia. The relatedness threshold of $IBD \geq 0.25$ is not necessarily a hard threshold for importation. As we do not have data on travel history from the publicly available WGS data we can therefore not know if some of the higher related individuals (e.g., $IBD \geq 0.25$) have a history of travel to Zanzibar or any other region in East Africa and acquired the infection there (and therefore show higher relatedness). We can therefore not exclude recent long-distance exchange. We have slightly rephrased this (Page 35, line 749-752):

"We identified a substantial number of infection pairs at the relatedness level of half-siblings or more ($IBD \geq 0.25$), **potentially** likely representing imported cases and providing evidence for recent, and most likely ongoing, genetic exchange between Zanzibar and the mainland".

62) On line 720, the claim that lack of *pfk13* mutations confirm that parasites remain sensitive to ASAQ is rather strong. Perhaps the word "confirming" could be changed to "suggesting" or similar?

RESPONSE: We agree with the reviewer and have toned this down. The Discussion section now reads (Page 37, lines 810-812):

"Our study did not identify any known *pfk13* artemisinin resistance-conferring mutations, **suggesting** confirming that parasites in Zanzibar remain sensitive to the current first-line treatment ASAQ."

63) It would have been fascinating to know whether some of the results presented here would have held if whole genome sequencing (WGS) had been used, e.g. would IBD

values between 0.75 and 0.9 still be seen, and would IBD ≥ 0.25 still be seen with each African country. Have the authors considered WGS, either themselves or in collaboration with others such as the MalariaGEN network? It might be useful to comment in the Discussion on what limitations the study has from having only 35 markers, and to what extent these might be addressed by WGS.

RESPONSE: We agree that a comparison of our amplicon panel to WGS would be interesting. We have not conducted WGS, and currently the protocol for selective whole genome amplification (as needed for DBS samples) is not established in our lab. In our study, a lot of samples were of very low parasite densities and might not yield sufficient coverage even after sWGA.

We will be happy to collaborate with others to work on WGS but would consider this a separate study.

It should be noted, however, that assessing genetic relatedness between polyclonal infections becomes more complicated with WGS data, both conceptually and methodologically. Obtaining phased genotypes from WGS data requires the use of statistical methods that are computationally intensive and may have limited accuracy, particularly when more than 2 clones are present. Most studies using WGS data have either attempted to infer a “dominant strain” from polyclonal infections using within host allele frequencies or have excluded polyclonal infections from the analysis altogether. Currently available methods based on IBD for *Plasmodium* spp. were developed for monoclonal infections or are adapted from human genetics: *hmmIBD* is designed for monoclonal infections (Schaffner et al. 2018); *isoRelate* is able to accommodate polyclonal infections (Henden et al. 2018) but is limited to biallelic loci as it is based on a diploid model. Thus, its applicability to infections with MOI >2 is unclear. Most infections in our study were polyclonal. Hence, with WGS data we would potentially grossly underutilize data or even introduce bias to the analysis due to informative missingness.

64) There are supplementary tables named S1-S5 in both Supplementary Results and Supplementary Methods which is confusing. Could these two supplementary documents be combined and tables renumbered accordingly?

RESPONSE: We can certainly do this, but ultimately this decision lies with the Editor. We have merged the Tables from Supplementary Methods with the Supplementary Results files. Tables have been renumbered accordingly and a Table of Contents has been added. The file is now referred to as “Supplementary Information”.

65) It might be useful to comment on why a different bioinformatics pipeline was used for the amplicon data (HaplotypR) vs the analysis of WGS data (bwa/samtools)

RESPONSE: HaplotypR is specifically designed for full length amplicon fragments, i.e., amplicon fragments are not sheared during sequencing library preparation (as is

the case for WGS). Furthermore, the sequence reads are not aligned to a reference genome, as the sequence reads of a marker are already aligned to each other after removal of the primer sequence. Importantly, sequence reads from a highly polymorphic region in some cases cannot be aligned to a reference sequence with standard alignment software due to too many mismatches to the reference sequence. During library preparation for WGS, DNA is sheared and consequently does not contain the primer sequence in each fragment, which is needed for demultiplexing the markers in HaplotypR. Thus, the resulting sequence data cannot be used for analysis with HaplotypR.

REVIEWERS' COMMENTS

Reviewer #1 (Remarks to the Author):

Holschuh et al have submitted a revised manuscript describing the development and application of a 35-locus amplicon panel, and its application to discern patterns of relatedness and movement within a set of nearby islands within Zanzibar. The resubmission has made substantial improvements to the manuscript with regards to the power of the panel and comparisons with prior work in other groups. I am happy to say the authors have answered all of my previous queries and I'm happy to recommend this for publication. Congratulations on generating a very nice study and a useful tool for the community.

Reviewer #2 (Remarks to the Author):

The authors have thoroughly responded to suggestions and queries both within the response letter and by implementing changes to the manuscript.

Only regarding Fig. 7 I suggest it could still be made more clear which connections colored by IBD metrics are deemed significant by permutation testing. Looking at figure and legend alone, it is hard to perceive. Something like mentioning that significance is only the case when specifically described, generally when using broader node membership classifications that generate larger sample sizes (?), might suffice. Also I was wondering if there is any meaning to the slightly differently sized nodes (circles) in the map?

Reviewer #3 (Remarks to the Author):

The authors have addressed all of my previous concerns.

Richard Pearson, Wellcome Sanger Institute

6 June 2023

Point-by-point responses to reviewer comments

Holzschuh et al., “Multiplexed ddPCR-amplicon sequencing reveals isolated *Plasmodium falciparum* populations amenable to local elimination in Zanzibar, Tanzania”

Reviewer #1 (Remarks to the Author):

Holzschuh et al have submitted a revised manuscript describing the development and application of a 35-locus amplicon panel, and its application to discern patterns of relatedness and movement within a set of nearby islands within Zanzibar. The resubmission has made substantial improvements to the manuscript with regards to the power of the panel and comparisons with prior work in other groups. I am happy to say the authors have answered all of my previous queries and I’m happy to recommend this for publication.

Congratulations on generating a very nice study and a useful tool for the community.

RESPONSE: Thank you for the kind and supportive words.

Reviewer #2 (Remarks to the Author):

The authors have thoroughly responded to suggestions and queries both within the response letter and by implementing changes to the manuscript.

Only regarding Fig. 7 I suggest it could still be made more clear which connections colored by IBD metrics are deemed significant by permutation testing. Looking at figure and legend alone, it is hard to perceive. Something like mentioning that significance is only the case when specifically described, generally when using broader node membership classifications that generate larger sample sizes (?), might suffice. Also I was wondering if there is any meaning to the slightly differently sized nodes (circles) in the map?

RESPONSE: We have added a sentence regarding the significance of connections in Figure 7. No connection was statistically significant by permutation testing and a larger sample size would likely be needed to achieve significance. The slightly differently sized nodes were just arbitrarily chosen as Zanzibar is rather small on the map. We have changed the node sizes so that they are all the same size.

“Significance between nodes was assessed using one-sided permutation test (100,000 permutations). Note, no connection was deemed statistically significant ($P < 0.05$) by permutation testing, likely due to small sample size.”

Reviewer #3 (Remarks to the Author):

The authors have addressed all of my previous concerns.

Richard Pearson, Wellcome Sanger Institute

RESPONSE: Thank you!